# Patients’ UX Impact on Medication Adherence in Czech Pilot Study for Chronically Ill

**DOI:** 10.3390/bs14060489

**Published:** 2024-06-09

**Authors:** Ondrej Gergely, Romana Mazalová, Michal Štýbnar, Antonín Hlavinka, Nicola Goodfellow, Michael Scott, Glenda Fleming, Leona Jochmannová, Ladislav Stanke

**Affiliations:** 1Department of Psychology, Faculty of Arts, Palacký University Olomouc, Křížkovského 10, 77900 Olomouc, Czech Republic; ondrej.gergely@upol.cz (O.G.); leona.jochmannova@upol.cz (L.J.); 2Czech National eHealth Center, University Hospital Olomouc, Zdravotníků 248/7, 77900 Olomouc, Czech Republic; michal.stybnar@fnol.cz; 3Center for Digital Health, Palacký University Olomouc, 77900 Olomouc, Czech Republic; antonin.hlavinka@fnol.cz; 4Medicines Optimisation Innovation Centre (MOIC), Pine House, Antrim Area Hospital Site, Bush Road, Antrim BT41 2RL, UK; nicola.goodfellow@northerntrust.hscni.net (N.G.); drmichael.scott@northerntrust.hscni.net (M.S.); glenda.fleming@northerntrust.hscni.net (G.F.)

**Keywords:** UX, medication adherence, eHealth, mHealth application

## Abstract

This article presents a comprehensive and multistage approach to the development of the user experience (UX) for an mHealth application targeting older adult patients with chronic diseases, specifically chronic heart failure and chronic obstructive pulmonary disease. The study adopts a mixed methods approach, incorporating both quantitative and qualitative components. The underlying hypothesis posits that baseline medicine adherence knowledge (measured by the MARS questionnaire), beliefs about medicines (measured by the BMQ questionnaire), and level of user experience (measured by the SUS and UEQ questionnaires) act as predictors of adherence change after a period of usage of the mHealth application. However, contrary to our expectations, the results did not demonstrate the anticipated relationship between the variables examined. Nevertheless, the qualitative component of the research revealed that patients, in general, expressed satisfaction with the application. It is important to note that the pilot testing phase revealed a notable prevalence of technical issues, which may have influenced participants’ perception of the overall UX. These findings contribute to the understanding of UX development in the context of mHealth applications for older adults with chronic diseases and emphasise the importance of addressing technical challenges to enhance user satisfaction and engagement.

## 1. Introduction

The present study aimed to investigate the relationship among baseline self-reported adherence to pharmacotherapy, as measured by the Medication Adherence Report Scale (MARS) questionnaire; the combination of beliefs about medicines (assessed through the Beliefs about Medicines Questionnaire, BMQ); and the level of user experience (measured by the User Experience Questionnaire, UEQ, and the System Usability Scale, SUS) with an mHealth application (Medimonitor) specifically designed and developed by the University Hospital Olomouc for older individuals with chronic heart failure (HF) and chronic obstructive pulmonary disease (COPD). The objective of this study was to determine whether these factors can serve as predictive indicators of changes in self-reported adherence, as measured by the MARS questionnaire, following the utilisation of the mHealth application. By examining the interaction among adherence, beliefs about medicines, user experience, and the impact of the disease-specific mHealth application, this study seeks to provide insight into the potential role of these factors in influencing medication adherence outcomes among older individuals with chronic heart failure and COPD.

Previous studies suggest that there is a relationship between usability and user experience with eHealth applications and patient adherence to medication. It was found that patients considered the Med Assist application easy to use and user-friendly, which may improve medication adherence [1]. There has also been variability in the usability and workload of different electronic medication adherence products among older adults, caregivers, and clinicians [2]. Usability, accessibility, reliability, versatility, and user experience are essential human-centric issues for eHealth applications that need further attention from researchers and practitioners. In general, these articles highlight the importance of considering usability and user experience in the development and deployment of eHealth applications to improve patient adherence to medication [3].

### 1.1. Chronic Heart Failure

Chronic heart failure (HF) is a complex clinical syndrome caused by a structural and/or functional abnormality of the heart and is characterised by shortness of breath, fatigue, and oedema [4]. This progressive health condition results in a lower quality of life and is a burden on the healthcare system [5].

The prevalence of HF in the adult population of industrialised countries is approximately 1 to 2%, and HF is more common in men [6,7]. With increasing age (over 70 years), the risk increases to a value exceeding 10% [7]. Patients with HF often undergo recurrent hospitalisations [8], which could be avoided if patients take more responsibility for their disease, are adequately educated, or are better monitored [9]. Telemedicine has the potential to significantly improve and support both patients and clinicians [10]. It can be helpful in optimising HF management, and, in part, in the prevention of acute HF [11].

Improving patient adherence is one of the main goals of telemonitoring in the diagnosis of chronic heart failure. Telemedicine interventions in patients with heart failure could significantly increase adherence to medications [12]. Furthermore, the findings suggest that mobile applications could improve adherence to cardiac rehabilitation [13]. There is also evidence of a positive effect of telemonitoring on weight control compliance in HF patients [10] or improved medication adherence using an Internet of Medical Things (IoMT) platform [14].

Consequently, the important point is that users of mHealth applications must be satisfied with their design and usability. An e-counselling platform with a professional and user-friendly layout may be helpful in the patient management of HF [15]. A well-accepted platform also helps patients feel confident while using it, and it could be easily incorporated into their daily life [14].

On the other hand, there are also negative impacts such as information overload for patients with HF or increased workload for healthcare professionals [16], as well as the increased use of healthcare resources without improving the clinical condition of heart failure patients [17].

In summary, telemedicine applications (e.g., mHealth applications) have the potential to support both patients and clinicians in the management of HF, but additional research is needed to determine the best practices and strategies to achieve optimal healthcare results.

### 1.2. Chronic Obstructive Pulmonary Disease

Chronic obstructive pulmonary disease (COPD) is a clinical state characterised by a limitation of airflow that is not completely reversible. COPD includes chronic cough, shortness of breath, and sputum production. The diagnosis is confirmed by spirometry [18]. In the Czech Republic, approximately 250,000 patients are diagnosed with COPD, representing almost 2.5% of the population. However, the estimated prevalence is 7–8% [19].

Telemedicine interventions for patients with COPD have the potential to improve their health status and quality of life, by preserving care in the home environment (between physical consultations) and supporting self-management [20]. Patients monitored by the telemedicine system have recovered faster from episodes of worsening symptoms and had a lower number of hospitalisations [21]. Telemedicine may also address treatment adherence to non-invasive ventilation [22]. However, the impact of telemedicine on preventing worsening symptoms and reducing the number of hospitalisations has not yet been definitively examined [23].

Technology acceptance is a key factor in the use of telemedicine among older people [24]. Usability testing in telemedicine applications is needed due to the different technological experiences of COPD patients [25]. The role of user experience in the management of COPD by older people is emphasised [26]. A systematic review showed that there are both positive and negative experiences [27]. In general, patients are more positive about the use of telemedicine than healthcare professionals are, who may view it as an addition to standard care. Users (patients or clinicians) can view telehealth either as a burden that generates dependence or as an empowerment that promotes self-care [27].

Telemonitoring can improve the quality of healthcare provided, the quality of life, and the prognosis of chronically ill patients, while reducing costs [8]. However, it is important to note the heterogeneity of different design options, technologies, and devices (e.g., presence or absence of human intervention), which makes it impossible to draw definitive conclusions about the benefits of telemonitoring.

### 1.3. Medication Adherence and Beliefs about Medication

Patient medication adherence is an integral prerequisite for successful treatment. It is characterised as a rational decision-making process that is driven by patients’ perceptions and experiences of the disease and treatment. It derives from the relationship between the perceived need to take medication and concerns about side effects [28]. In developed countries, only half of the chronically ill patients adhere to treatment [29,30]. The main reasons for low adherence (or non-adherence) are a lack of information about the disease or treatment, taking too many medications at once (complexity of the treatment regimen), and forgetfulness [31]. Adherence to chronic disease management is critical for achieving better health outcomes, quality of life, and cost-effective health care [32]. Beliefs about medication are related to adherence and can predict self-reported adherence better than gender, education, or the number of prescribed drugs [33].

#### 1.3.1. The Beliefs about Medicines Questionnaire (BMQ)

The beliefs about medicines questionnaire (BMQ) [34] is a tool designed to assess patients’ beliefs and perceptions regarding medicines. The BMQ is designed to capture various aspects of patients’ perception on medicines, including both the general views and specific concerns related to the drugs they are personally using. BMQ consists of the following two sections: BMQ-General and BMQ-Specific. The general section assesses beliefs about medication in general, while the latter assesses representations of medications prescribed specifically for the personal use of the patient. The questionnaire version used within the described study consisted of 12 items for assessing the general views and 11 items for personal views.

BMQ-General further assesses general beliefs about medicines through two subscales:General-Harm: Measures beliefs about the potential harm caused by medical drugs.General-Overuse: Assesses beliefs about the over-administration of medical drugs by physicians.

BMQ-Specific evaluates specific beliefs about medical drugs prescribed to the specific person. It also comprises two subscales:Specific-Concerns: Focuses on concerns about medical drugs prescribed for the specific person and the diseases the person suffers.Specific-Necessity: Examines beliefs about the necessity of personal medication.

The Necessity–Concerns Differential (NCD) provides numerical value of benefit (Necessity) against perceived risk (Concerns) perception. NCD is an indication of the relative importance of these beliefs among different individuals. It might be understood as implicit cost–benefit analysis (CBA), where cost is equal to concerns and benefit to necessity. Positive values indicate that the necessity (benefit) scores are higher than concerns and vice versa.

The Czech version of the Beliefs about Medicines Questionnaire (BMQ-CZ) has been validated in a sample of patients with arterial hypertension, diabetes, and rheumatological diseases. Its structure and internal consistency were confirmed [30]. The psychometric properties of BMQ-CZ were further validated, with factor analysis supporting its four-factor structure. Concurrent validity demonstrated that BMQ-CZ possesses sufficient psychometric properties, making it a reliable self-assessment tool for measuring beliefs about medication [35].

However, there are challenges in the BMQ application for complex conditions. It is recommended to develop condition-specific versions as the current version might lead to misreading by the participants [36]. There were even concerns raised about the BMQ’s ability to represent beliefs about medical drug underuse. Nowadays, the BMQ has been developed and there are several disease-specific versions containing treatment-specific items.

#### 1.3.2. The Medication Adherence Report Scale (MARS)

The Medication Adherence Report Scale (MARS) [37] is a self-report questionnaire designed to assess patients’ adherence to various medication regimens. It was originally developed as the MARS-10, consisting of 10 items aimed at eliciting patients’ reports of nonadherence. The MARS-5 is a shorter version of the original tool, comprising five items that describe a range of nonadherent behaviours, on which patients are asked to rate the frequency, in their own perspective, of each adherence-related behaviour on a five-point scale.

The assessment of the psychometric properties of the Czech version of the MARS questionnaire demonstrated that it is an acceptable tool to measure adherence. The MARS-CZ showed satisfactory internal consistency (Cronbach’s α = 0.54), strong test–retest reliability (r = 0.83, *p* < 0.001), an intraclass correlation coefficient (ICC) of 0.63 (95% confidence interval), and a positive correlation (r = 0.62, *p* < 0.001) with the eight-item Morisky Medication Adherence Scale [38].

### 1.4. User Experience, Technology Acceptance, and its Assessment in the Context of Healthcare

Scientific work in the field of healthcare is very complex. It faces many challenges and ethical considerations. Each person is unique and may respond differently to treatment or health interventions. Researchers must consider various factors, including age, gender, genetics, lifestyle, and other variables, which complicate the interpretation of the results and the formulation of clear conclusions. The combination of these factors makes health research challenging, requiring ample planning, transparency, and respect for human rights.

It is important to have tools that help to objectively evaluate the user experience of health application technology solutions. The optimal approach to user experience (UX) evaluation involves a combination of both qualitative (user testing) and quantitative methods (SUS, UEQ). Obtaining comprehensive data is essential for improving systems designed to actively support adherence. This is because different patients may appraise the same application differently, due in part to their digital skills, but perhaps also because of their different needs (a chronically ill patient has different needs than a person without chronic illness). User experience evaluation is therefore a comprehensive concept of all possible user reactions regarding a specific application [39].

#### 1.4.1. Qualitative Research in Medical Application Testing

Usability testing is a qualitative method that focuses on understanding the user experience and behaviour when interacting with technology. The method uses scenarios, i.e., descriptions of situations that are intended to be as close as possible to the real situation as users might face it. The researchers observe how the respondent handles the task. Researchers then receive specific feedback, revealing the most common crisis moments (when the user does not know how to proceed) and the strengths of the application (those features that the users understand).

#### 1.4.2. SUS: System Usability Scale

The SUS was developed in response to the need to find a quick and efficient scale for usability assessment. The SUS has ten items rated on a five-point Likert scale (strongly disagree—strongly agree). Respondents should be presented with this scale soon after exposure to the application and asked to record their immediate reaction to each item (without thinking too long) [40]. The responses are numerically processed to give an absolute score that indicates the system’s overall usability.

In the set of key psychometric properties of the SUS examined, based on research conducted with the standard English version, estimates of internal reliability using Cronbach’s α ranged from 0.83 to 0.97, with a mean of 0.91. Estimates of concurrent validity showed significant correlations ranging from 0.22 to 0.96. The correlations of SUS with other methods of subjective usability assessment ranged from 0.50 to 0.96, while the correlations with objective metrics of task completion (success rate) ranged from 0.22 to 0.50 [41]. The SUS has been translated into a number of languages. The average reliability estimate in these language versions was approximately 0.81 (Cronbach’s α), which is slightly lower than that which is usually presented for the English version, but still significantly higher than the standard minimum criterion of 0.70. Estimates of concurrent validity with various other indicators of perceived usability presented significant correlations ranging from 0.45 to 0.95 [41].

#### 1.4.3. UEQ: User Experience Questionnaire

The UEQ is a comprehensive quantitative method for measuring user experience. The original German UEQ was created in 2005. After several studies and modifications, from the original 229 potential items, the method settled on 6 scales with 26 items that measure various aspects of user experience [42]. These include the following:Attractiveness: How do users like or dislike the application?Perspicuity: Is the application easy to use?Efficiency: Can users meet the goals of the application without effort?Dependability: Is the application reliable?Stimulation: Is the application interesting? Does the user want to use it?Novelty: Is the application innovative? Is it attractive in some way? Is it needed?

The results of the evaluation of the reliability and validity of the UEQ showed that the scales were not independent of each other. Combining perspicuity, efficiency, and dependability into pragmatic aspects, as well as novelty and stimulation into hedonic aspects of UX significantly improved model fit. The systematic variations of product properties and their correlations with the System Usability Scale (SUS) supported the validity of these two factors [43].

#### 1.4.4. ISO 14915 and 9241

To design and consequently assess the mHealth application presented in this paper, ISO 14915 and ISO 9241 were employed [44,45]. The ISO 14915 standard provides essential guidelines and recommendations for the design of ergonomic multimedia user interfaces. It establishes a comprehensive framework for addressing the complexities and considerations involved in designing user interfaces for applications that incorporate different media elements. The standard focus on the ergonomic design of multimedia applications is primarily intended for professional use.

One of the primary objectives of ISO 14915-1 is to enhance the effectiveness, efficiency, and user satisfaction in operating various applications. The principles of ergonomic design play a crucial role in achieving this goal by considering user characteristics, task requirements, and the environmental context of system usage. By incorporating ergonomic design principles, multimedia interfaces can be customised to improve usability, accessibility, and overall user experience. The wide range of media types and their interactions in multimedia interfaces introduce various perceptual, cognitive, and ergonomic implications for users. Factors such as high perceptual load, structural, and semantic complexity, and the need to convey substantial information pose unique challenges.

ISO 14915 provides a set of requirements and recommendations that specifically address the ergonomic design of multimedia software-user interfaces. User testing and assessment play a crucial role in evaluating and refining multimedia interfaces. While ISO 14915 does not explicitly cover user testing methodologies, it aligns with the general ergonomic principles outlined in ISO 9241-10. These principles encompass aspects such as suitability for the task, self-descriptiveness, controllability, conformity with user expectations, error tolerance, suitability for individualisation, and suitability for learning. Applying these principles ensures that multimedia interfaces meet user expectations, facilitate ease of use, and accommodate individual user preferences.

## 2. Materials and Methods

This article applies a mixed methods research design. The methods are described in chronological order to show how they were applied throughout the development of the application. Development and testing were divided into the following consecutive and logically constructed steps (Figure 1):Step 1—Preliminary qualitative usability and UX research: Application wireframes and high-resolution prototype assessment.Step 2—Follow-up qualitative usability and UX research: Application working prototype testing.Step 3—Small-scale testing in the patient environment.Step 4—Pilot study with patients with chronic heart failure (HF) and chronic obstructive pulmonary disease (COPD).Step 5—After the pilot follow-up, qualitative research is conducted with the pilot participants.

### 2.1. Step-by-Step Procedure of the Application Development

This section provides a concise overview of the research process, outlining its five steps and their interconnections. Each step is described in its own section, following a consistent structure (if applicable): Specific Objective, Methods and Instruments, Scenario/Task, Participants, Implementation/Execution, Results, and Discussion and Integration with Next Step.

#### 2.1.1. Preliminary Qualitative Usability and UX Research—Application Wireframes and High-Resolution Prototypes

The preliminary qualitative study consists of two main parts—wireframe testing and high-resolution prototype testing. The results of the wireframe testing are directly used for the preparation of high-resolution prototypes. The contents of both parts are described in more detail in the sections below.

**Wireframes:** Wireframes are simple graphical sketches that are used for the basic visualisation of a proposed application and the layout of its elements. Aesthetics are not a factor in wireframes; their main purpose is to define the structure and verify how such a structure affects the user.**Specific Objective:** The objective of testing the wireframe model was to conduct in-depth interviews and test user experience on a group of older adults with minimal digital literacy.**Methods and instruments:** The assessment of wireframes was conducted through in-depth interviews. The data were then analysed using qualitative analysis techniques.**Scenario/Task:** To verify the feasibility of the tasks, the following features were tested: requesting a prescription for medication, making an appointment with a doctor, recording measurements, and setting medication reminders. The usability of the browsing application was verified by performing tasks such as “returning to the home screen” and “navigating across sections without using a menu”. The evaluation of the design patterns included the following: a dashboard with tiles, a list/detailed layout, tips displayed in the detailed view with options to collapse/expand and view more information, distraction-free task interactions within pop-up/modal dialogues, and overlays with guided instructions.**Participants:** A total of 5 respondents participated in the application’s wireframe testing. The group consisted of 4 women and 1 man, with a mean age of 71 years. None of the participants had previous experience with digital medicine.**Implementation/Execution:** The testing process started without using the guide so as to assess the intuitiveness of the interface, with the guide being tested at the end of the evaluation. The aims of the testing were as follows: to verify the feasibility of the tasks; to verify the usability of the browsing application; and to evaluate the design patterns used.
**High-Resolution Prototype Testing:**
**Specific Objective:** To test the graphical clarity of the application and find the most suitable display orientation (portrait or landscape).**Methods and instruments:** The data were analysed through a qualitative analysis based on short transcripts of interviews with end users. **Scenario/Task:** The following user testing scenarios were used: taking measurements with the instrument, taking one or more medications, requesting a prescription, conversation with a doctor, and card sorting.**Participants:** Older adults (future users) and 4 medical staff members of the cardiology department, including 2 doctors and 2 nurses.**Implementation/Execution:** For the second user test, we prepared the first sketches of the application in Figma and presented them to two groups of participants.
**Results of Wireframes and High-Resolution Prototypes Testing:**
The data were analysed and grouped on similar topics, leading to the following recommendations:It was suggested that the medication list should be visualised in a manner resembling the agenda for today’s measurements.

To enhance the medication request feature, it was recommended that the text field be replaced with a multi-select option, allowing users to choose from a list of available medications. Additionally, including an “other—please specify” option can add flexibility. It is also advisable to leave out the physician’s note and provide clear information on how the ePrescription will be delivered to the patient on the final screen.

Regarding the request for a consultation, the implementation of a multistep dialogue can help patients focus on each item of the form. Instead of providing a subject name, users can select the reason from a predefined list. To improve the user experience, it was recommended that the available free time slots should be displayed on the calendar, similar to common COVID-19 tests and vaccination reservation systems. However, this enhancement requires updating the Hospital Management System’s (HMS) scheduling capabilities. Furthermore, after submitting a video consultation request, clearer information should be provided on how patients were notified of the doctor’s acceptance and the selected time slot. Users expressed confusion as to whether the ordering system was intended for personal or online (video conferencing) visits. However, for the initial release, only video conference visits were supported. Clarifying the call initiation process was also recommended, as users were uncertain about which side should initiate the call.

Regarding navigation through the application, test users encountered difficulties re-turning to the home screen, possibly because they had overlooked the tabs in the headings.

**Discussion and Integration with Next Step:** The results of the wireframe testing were directly used for the preparation of the high-resolution prototypes. The completed reports and the collated findings, including recommendations, were presented to the technical experts responsible for the development (designers, programmers, etc.). The identified recommendations offered valuable information to enhance the usability and user experience of the mobile application. Addressing these issues can lead to a more intuitive and user-friendly interface, ultimately improving user engagement and satisfaction.

Wireframe model testing highlighted, among other things, the importance of correct button naming. Figure 2 and Figure 3 show the first and second versions of the wireframe prototypes, respectively, illustrating the wireframes before and after the user feedback was collected.

#### 2.1.2. Follow-Up Qualitative Usability and UX Research—A Working Prototype of the Application, Hands-on Testing

Hands-on testing of the application’s working prototype was performed to collect feedback from end users and evaluate the performance of the digital solution in the actual pilot setting. 

**Specific Objective:** The aim was to train end users to use a close-to-final version of the user-facing digital solutions to be deployed in the pilot study of chronic heart failure and COPD use cases, provide end users with the opportunity to challenge their functionalities and user-friendliness, and then for them to provide feedback indicating what changes still needed to be made.**Methods and instruments:** The end users were presented with tools that had been developed and improved during the previous stage. Participants accessed the prototype through a tablet owned by the organisation. Feedback was collected at different times during this development stage using a variety of different methods. Using feedback collected during interviews with participants, both use cases were evaluated using the application of ISO Standards for multimedia design (ISO 14915 and ISO 9241). General feedback was provided first, followed by specific feedback and recommendations for each mock-up screen. A concurrent ‘think out loud’ approach was used to collect reactions to the application and identify areas requiring particular attention during the demonstration of the application and user training. Participants were encouraged to verbally express their reactions, thoughts, feelings, and opinions about the prototype throughout their engagement with the researchers. Researchers took notes and recorded the sessions to capture feedback accurately.**Scenario/Task:** End users were asked to perform several tasks reflecting typical use case scenarios like ordering medication, undergoing video consultations, and conducting measurements. Hands-on experiments were conducted with two specific groups, namely the patients and healthcare professionals. The steps and tasks involved included the following steps: assessing the perception of colours, accessing the prototype, navigating to different features of the ‘Dashboard’, navigating to ‘Dashboard’ from within the application, view and enter the measured biomedical signals (blood pressure, heart rate, oxygen saturation, etc.), view the medication list, and view the ‘daily to-do list’.**Participants:** In the first round of testing, five respondents participated—4 women and 1 man. This group included patients and healthcare professionals. After the initial 5 respondents were evaluated, another round of testing was organised and conducted by psychology students, with 33 respondents participating in this second round.**Implementation/Execution:** The pace of the session was determined by the participants. After the demonstrations, the participants were encouraged to use the application while the researcher was still present and available to answer questions and troubleshoot any issues. There was a break to allow participants to rest, process the information provided, and familiarise themselves with the application. The user training manual included an example task that the participant could complete, containing mock data to enter as readings into the application. When the second session started, the participants were asked how they fared with completing the tasks and if they had any questions or queries about using the application. Feedback was then collected about the application, as described below. If a participant discovered a serious ‘bug’ in the application (which could potentially be a crisis moment for all other users), the ‘bug’ was promptly fixed, and the next respondent proceeded to work with the modified version.When the participants returned for the second session, they were first asked to complete a short activity under observation, i.e., a moderated test. During the test, the participant was asked to ‘think out loud’ and thus point out ‘stumbling blocks’. The facial expressions and movements of the user were observed and noted, and the observer documented how often the participant asked for help and if there were specific points where many users needed support. It was observed whether the buttons or items on a touchscreen were considered easy to use and whether the response to the application corresponded to what was expected by the user.
**Results:**
**Interim Findings from Student-Led Interviews:** Subjectively, respondents preferred the colour version (in 66.7% of the cases) and rated it as readable and cheerful. The dark version came in second place (in 24.2% of cases) and was described as distinctive and logically organised. The light version received the fewest votes (9.1% of the time) and was rated as monotonous, bland, or amateurish.**Home Screen Testing:** To optimise user interaction and navigation, it was recommended that the navigation bar be moved from the left side to the bottom of the screen. This alteration was rooted in the perception that the previous navigation bar resembled tiles, prompting a redesign from a table layout to a tile layout. The revised navigation bar should include a concise selection of three to five essential items, including “home”, “settings”, and “help”. To enhance user comprehension of the extended content on the homepage, the introduction of a “Scroll below” button was proposed. This addition was aimed at providing a clear mechanism for users to access additional tiles present below the initial view. The activity screens are shown in Figure 4.

**Figure 4 behavsci-14-00489-f004:**
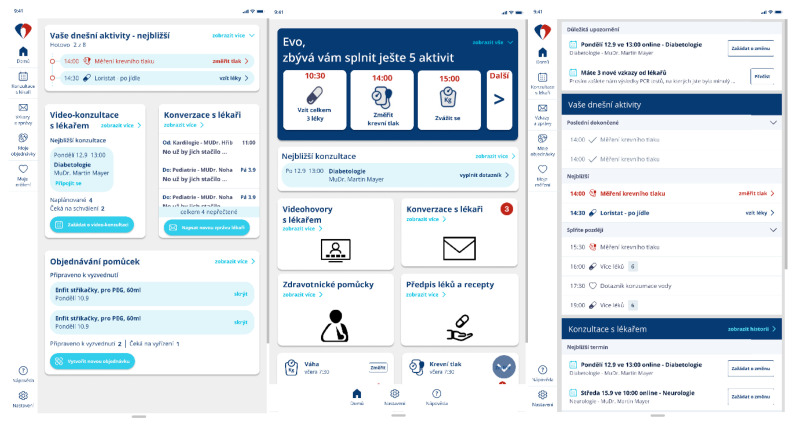
Activity screens (from top left to bottom left: Today’s activities, Video consultations with doctors, Conversations with doctors, Ordering supplies, from top middle to bottom middle: Today’s activities, Video consultations with doctors, Conversations with doctors, Medical supplies, Prescriptions, from top right to bottom right: Important notifications, Today’s activities, Upcoming consultations).

**Medication Order Testing**: The enhancement of logic within the system involved several key considerations. Older adults were observed to have a greater comprehension of pop-up windows, thus advocating the integration of such windows into the user interface. Additionally, optimising the user experience involved strategic modifications, such as adjusting the amount of packaging options available on the medication selection page.

An imperative adjustment involved the placement of the message to the doctor within the ordering process. To facilitate a seamless user journey, it was suggested that this message be positioned after the ordering process. Alternatively, a user-friendly approach could involve the introduction of a pop-up window that offers a dedicated “send order” button. This pop-up window could additionally incorporate a feature enabling users to “add a note”.

A progressive evolution from the initial concept involved refining the user-interaction model. The initial idea of entering text while selecting a drug or medical aid, denoted as “click and start typing the name of the drug, then select from the list”, was proposed to be replaced by a more intuitive interface. In this proposed interface, the concept of an “adding drug” button was introduced, using a pop-up window to streamline the process of selecting medication or medical aids. This design iteration was aimed at enhancing user engagement and comprehension.

The overarching operational principle guiding the button functionality within the application was “select–complete–send”. This approach seeks to streamline user interactions by providing a consistent and user-friendly framework. If this approach proves efficacious and well-received by users, it is recommended that this principle be harmonised across the entirety of the application’s interface, ensuring a coherent and intuitive user experience.

Central to all the orders placed within the system was the inclusion of a “message from the doctor”. Recognising the multifaceted information needs of patients, this component extended beyond simple updates on order status. In addition to being informed of the approval or rejection status of an order, patients also anticipated receiving contextual information on the implications of these statuses.

For example, an approved order will incorporate the following informative directive: “Please wait for confirmation from your doctor. Subsequently, an SMS message containing an electronic prescription will be sent to you”. This comprehensive communication ensured that patients were not only informed of their order’s approval but also aware of the ensuing steps.

Conversely, in instances where an order faced rejection, an interactive text box was integrated, which gave the attending physician the means to provide a rationale for the decision. This interactive mechanism fosters direct communication between the patient and the healthcare professional, enabling an informed understanding of the reasons behind the rejection of orders.

By incorporating these strategic communication elements into the order management process, the application aimed to empower patients with clarity, actionable information, and a sense of participation in their healthcare journey.

**Testing of Medication Confirmation:** Part of the medication confirmation test involved the introduction of an additional column that allowed users to defer medication intake to a later time. This feature offered users the option of postponing medication consumption, improving user autonomy and accommodating their diverse schedules.

When the option “take the medication later” was selected, a warning message was sent. This prompt presented users with distinct options, differentiating between two courses of action as follows: “I will not take the medicine today only” and “I will never take the medicine again”. This duality of choices catered to varying circumstances and user preferences, fostering a tailored and considerate experience. The screen, which includes confirmation of medication consumption, is shown in Figure 5.

**Conclusions Based on ISO Standards Methodology:** In this section, conclusions and recommendations are presented based on ISO standards methodology, including homepage design suggestions, the results of the homepage assessment, and the results of the ‘Take pill’ task assessment using ISO 14915.**Homepage Assessment with the Use of ISO 14915:** The conclusions from the homepage assessment using ISO 14915 are as follows: the need to scroll is not well communicated; and an arrow or text of “show more” is needed. Simplification of the application while incorporating more pictograms is needed, as the current amount of text is too overwhelming.

Based on the conclusions, the following recommendations were made: Firstly, the first contact between the user and the application should be positive—the application should be organised well and easy to navigate. Secondly, the homepage will be redesigned into a new tile scheme with pictograms (effect of image priority and simplicity of understanding). Thirdly, the sidebar will be moved to the lower part of the screen and contain lesser information (home, settings).

**Take Pill Task:** The recommendations from the ‘take pill’ task assessment using ISO 14915 are as follows: Firstly, there is a necessary interaction with each drug in one of the three states, which is consistent and understandable. Secondly, for “take later” and “do not take” statuses, pop-up windows with specific information will pop up. For “I do not take” it will be the following: I do not take at all/remind later. For “remind later” the user chooses when he wants to be reminded. Thirdly, for each drug, it is enough to mark the status, and then only the final button is marked “close”, and it does not need to be confirmed.**Colour and Layout Perception Testing in a Broader Group of Participants:** Participants were given these three versions (Figure 6) each time in various orders to choose the best one. The most popular version was the third, i.e., the colour version. This version was voted for in 22 cases.

**Figure 6 behavsci-14-00489-f006:**
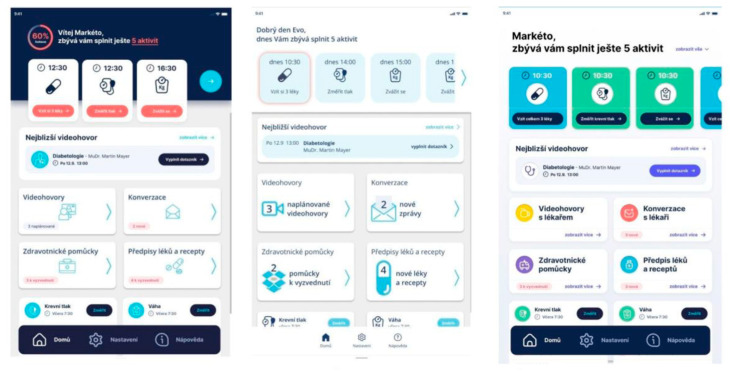
Colour variants of the home screen; from the left: dark, bright, and coloured version.

The respondents described the colour version of the Medimonitor application as clear, legible, cheerful, colourful, and understandable. The respondents positively assessed that the colours needed to be understood as well as the text. The colour differentiation of the individual sections facilitated orientation. A negative evaluation usually referred to the fact that it was too colourful, even childish.

The respondents perceived the dark version as interesting, logically arranged, dark, and expressive. Most of the time, they criticised the upper colour as being too dark.

The light version was too monotonous, dull, indistinct, amateurish, unfinished, clean, and unclear, according to the participants.

Respondents were given two colour versions to order medical aids below (Figure 7). The participants chose the dark version more often. The dark version was selected for 20 cases. The reasons for choosing the dark version were that it was clearer, and that the red had a stronger effect.

Following the colour palette evaluation, the layout and buttons were also assessed. The top placement of the order button was the most favoured, as it was selected by 18 respondents. The bottom placement was chosen by nine respondents, while three respondents did not find the button at all. It was recommended to enlarge and emphasise the button for ordering medical aid and to place it at the top.

**Discussion and Integration with Next Step:** User testing of the Medimonitor application design for patients older than 65 years was important for creating an effective mobile application that reflected the needs of the target group but was also applicable to other age groups. Testing of the mobile application, which was available on the web, took place at the same time as the telemedicine portal was being tested. The development of the mobile application consisted of the so-called “patient-centric” approach to create an intuitive, easy-to-use application from the point of view of UX/UI. Emphasis was placed on the appropriate medical colour scheme based on the psychology of colours and healthcare fonts, as well as on making decisions about important design elements and details. The mobile application was also created in such a way as to follow current trends. Testing of the application took place in cooperation with psychologists with an emphasis on qualitative data collection to provide a better understanding of what respondents thought. The latest version of the application is in light tones, which evoke the feel of a hospital environment in respondents, and it was redesigned from a line form to a tile design. The tile design with pictograms was best perceived by the respondents, and the individual colours increased the feeling of comfort. A stronger link to the application was created by addressing the patients by their name and a motivational scheme showing the percentage of tasks already completed. It was repeatedly necessary to test the correctness and meaning of the CTA buttons used in both the mobile and web forms. All the above recommendations were implemented in the existing solution and again tested.

#### 2.1.3. Small-Scale Testing in Patients’ Environment

During this intermediate phase, an examination of the Medimonitor application was performed using a concise but comprehensive small-scale testing regimen. 

**Specific Objective:** The primary objective of this phase was to establish the operational integrity and stability of the system, laying the foundation for its subsequent deployment in a pilot clinical study.**Methods and instruments:** A small-scale live demonstration was designed to collect feedback from end users and evaluate the performance of the digital solution and data transfer in a real-world environment before the pilot study was conducted. The demonstration also tested the methods and procedures planned to be used within the pilot study. A stable Wi-Fi connection at home, to be in good health condition, to be able to participate in the pilot, and to be a user of a smartphone or tablet were the requirements. Seamless data transmission was a crucial point of this examination. Device-generated readings were transmitted to the Medimonitor application using Bluetooth technology to the FNOL servers.**Scenario/Task:** Participants were asked to use the digital solution daily and perform several planned tasks that reflected typical use case scenarios, such as ordering medication or video consultations, performing vital measurements and completing questionnaires. These tasks included the capturing of vital parameters including weight, blood pressure, heart rate, oxygen saturation, ECG readings, and activity levels (HF group). Additionally, participants interacted with questionnaires, partook in video consultations, exercised medication ordering functionalities, and routinely accessed their medication lists, as well as a ‘daily to-do list’.**Participants:** For the application evaluation, four people from various age groups were selected as the sample. The participants recruited were not representative of the target user group but were able to use the solution at home for the duration of this development stage.**Implementation/Execution:** In parallel, a rigorous assessment of the user interface and experience was carried out. Feedback from participants played a pivotal role in refining the user app manual and the Medimonitor application interface. This iterative process ensured an intuitive and user-friendly environment. End users were presented with the application developed and improved during the development stages described above. If necessary, there was also the opportunity to adjust the solution before the start of the pilot study.**Results:** The inclusion of contextual information was found to greatly enhance participants’ comprehension and accuracy in executing complex tasks. This finding resonates particularly when dealing with multi-step processes, underscoring the importance of clarity in task interpretation. Furthermore, the visual presentation of tasks has considerable significance. A variant that prominently highlighted incomplete tasks in a conspicuous red frame at the top received approval from all participants. This visual cue was found to effectively guide the participants’ actions. Similarly, a consensus emerged in favour of using a green checkmark to signify the completion of a task, reflecting a well-established and universally understood symbol. Age-related nuances in visual preferences were also discerned, emphasising the importance of catering to the diverse demographic characteristics of users. The preference for a white button version among older participants, juxtaposed with the inclination toward a dark version among younger participants, highlighting the multifaceted nature of design considerations.**Discussion and Integration with Next Step:** In summary, this intermediate phase, conducted with attention given to these details of the UX and the stability and reliability of the application, was tested to push the designed Medimonitor application toward the clinical pilot study presented in the next step of the development process. Although several optimisation loops were performed, some minor bugs and glitches continued to occur. These limitations were mainly related to the stability of the application and of the Bluetooth connection that was used to communicate with the connected medical devices; however, it was not possible to postpone the clinical trial due to the project deadlines. Therefore, it was decided that back-end optimisation, with no effect on the GUI, would continue.

### 2.2. Pilot Study with Patients with Chronic Heart Failure (HF) and Chronic Obstructive Pulmonary Disease (COPD)

The study was a non-randomised, single-armed, cross-sectional prospective interventional pilot study with a qualitative interview component. The pilot study aimed to recruit 25 participants for the HF use case and 25 participants for the COPD use case. The intervention piloted in this study was a Medimonitor application aided by various medical devices, from which the data were collected by Medimonitor and further transferred to servers and analysed. Hence, the main objective was to monitor the health status of patients with chronic diseases, given the ability to contact patients when they needed an additional check-up at the hospital. 

The pilot study was carried out with service users and medical professionals in Olomouc, Czech Republic, University Hospital Olomouc (FNOL). The pilot was managed by researchers from FNOL. This study aimed to test the hypothesis that new digital solutions can provide opportunities for older adults and health professionals to support the management of chronic heart failure and COPD. If participants provided informed consent, they were included in the study, and continued with the Compliance Form. If the person did not fully complete the baseline questionnaires, the person was excluded from the study. If the participant did not fully follow the instructions or did not properly use the items, the person was excluded from the study.

In terms of the application assessment, the target sample size for both the HF and COPD use cases was 25 service user participants and up to two medical doctors and nurses. These sample sizes were selected to be as representative as possible to provide valid answers and to fall within the scope of available resources. There were two main user groups as follows:Target users: People older than 60 years of age with chronic heart failure or chronic obstructive pulmonary disease who lived at home. We aimed to recruit 25 participants for each use case.Health professionals: Medical doctors and nurses who work for the University Hospital Olomouc.

The eligibility criteria for patients with HF are described in Table 1.

Similarly, the eligibility criteria for COPD patients are described in Table 2.

Potential participants were selected from ambulatory patients at the University Hospital Olomouc according to the eligibility criteria described above. Participants were selected on the basis of convenient and random sampling. The sampling was convenient because the participants were available during their regular in-patient visits. If participants provided informed consent, they were included in the study and continued with the Compliance Form. If participants did not fully complete the baseline questionnaires, they were excluded from the study. If participants did not fully follow the instructions or did not use the items correctly, they were also excluded from the study.

A total of 25 patients per use case were selected to participate in the pilot study in Olomouc. Participants received an information sheet and privacy notice, and once eligibility was confirmed, a consent form was obtained. The age distribution of the participants for both groups is shown in Figure 8.

The age distributions for both groups appear to be relatively normal, with the COPD group showing a slightly negative skew and lower kurtosis (flatter distribution) compared to the HF group. This suggests that the ages of participants in both groups are relatively evenly distributed, although the COPD group may have a slightly younger peak.

In terms of age distribution, the COPD patients have a mean age of 68.7 years and a median age of 68.6 years. The distribution appears to be slightly negatively skewed, as indicated by a skewness value of −0.126 and a kurtosis of −1.160, implying lighter tails and a less peaked shape.

On the other hand, the HF group has a slightly higher mean age of 69.4 years, with a median age of 70.1 years. The age distribution for this group shows a slight positive skew with a value of 0.091, while the kurtosis is −0.194, indicating a relatively more normal distribution with fewer extreme tails.

These findings suggest that while both groups had similar mean ages, the age distribution had a slightly younger peak and a flatter shape compared to the distribution of the HF group, which was relatively closer to a normal distribution.

All participants received a face-to-face introduction to the Medimonitor application and training on how to use both the application and connected medical devices. The user manual contained detailed step-by-step supportive guidance and information and was provided to participants in a hardcopy format for reference during the pilot. Participants were asked to use the application and connected medical devices for 12 weeks.

The goals of these interventions were to help people self-monitor their health conditions and medication adherence in their own home. Participants diagnosed with chronic heart failure were provided with the following medical and well-being devices, together with the data gateway (tablet), for use at home:blood pressure monitor (Beurer BM54),smartwatch (Xiaomi Mi Band 6),ECG (Beurer ME90 ECG),weight scale (Beurer BF600),pulse oximeter (Beurer PO60),smart pillbox (developed in-house),tablet: Lenovo Tab M10 HD.

Participants diagnosed with COPD were provided with the following medical and well-being devices, together with the data gateway (tablet), for use at home:blood pressure monitor (Beurer BM54),spirometer (Spirobank),smart inhaler (FindAir),pulse ring oximeter (BodiMetrics Circul),tablet: Lenovo Tab M10 HD.

During a 12-week pilot initiative, both groups of participants, those with chronic heart failure (HF) and those diagnosed with chronic obstructive pulmonary disease (COPD) participated in comprehensive health monitoring. Both groups utilised the devices mentioned above to measure key health indicators daily. For the HF group, additional metrics such as weight (including body mass index and percentage of body fat), electrocardiogram (ECG), and daily physical activity were also recorded. In contrast, the COPD group focused on spirometry readings and inhaler usage. Adherence in the HF group was assessed using the smart pillbox.

These health data were wirelessly transmitted via Bluetooth to a tablet with a dedicated application. Through this application, participants could access their health metrics, manually input data, if necessary, communicate with healthcare providers, and schedule virtual consultations. Participants also completed daily health questionnaires and weekly medication use surveys. The prescribed medication regimen for each participant was integrated into the application and presented as a complete list of medications and a daily checklist for adherence.

Regular inquiries were made about any modifications to the treatment regimen during the week, allowing appropriate adjustments. If participants reported changes in their medication, the application was updated promptly to reflect these modifications. In this way, participants with both HF and COPD would benefit from personalised and proactive healthcare management through the integration of technology and patient engagement.

### 2.3. After the Pilot Follow-Up Qualitative Research with the Pilot Participants

The qualitative evaluation was aimed at exploring the experiences and perspectives of the patients who participated in our pilot study. Patients were contacted immediately after the end of the study, provided they gave their consent to capture their experience. We opted for mobile phone interviews, as participants had completed the study on different dates and at different times and had a doctor’s visit and medical examination at the end of the study. We asked them to evaluate their participation and were also interested in the perceived benefits of participating in the study or any technical problems related to the use of devices. This information may help us improve similar studies in the future.

Individual semi-structured interviews were conducted with 19 patients in the HF group (16 men, 3 women) and 20 patients in the COPD group (15 men, 5 women). The data were analysed using the inductive thematic analysis method [46].

## 3. Results

### 3.1. Pilot Study of Patients with Chronic Heart Failure and Chronic Obstructive Pulmonary Disease

First, it is important to mention that neither the smart pillbox (HF group) nor the smart inhalers (COPD group) provided usable information because the patients did not use them correctly or not often enough; therefore, they could not generate meaningful and statistically usable data. Most of the HF patients did not use the smart pillbox at all. To avoid influencing the adherence of the participants, it was not possible to inform them at the beginning of the pilot that the frequency of use of the pillbox was being measured. Instead, information was provided on how to include them to improve the design of the pillbox. It was a challenging task to monitor adherence, especially from an ethical perspective [47], which is even more pertinent in the age of digital medicine [48]. Participants were briefed about the true nature of the pillbox after the study. Most of the participants were not interested in improving the design of the pillbox, which was also an interesting finding. The underuse of smart inhalers was also an issue. This was especially problematic if the device was dismantled from the inhaler or the inhalers had different mechanical designs, leading to incompatibility issues. High failure rates were also mentioned, with recommendations for implementing higher quality controls [49], although their failure rates ranged from 13 to 16%.

The administration of the questionnaires throughout the pilot study yielded the following results (Figure 9). For the COPD group, the mean attractiveness score was 0.991, with a median of 1.000 and a standard deviation of 0.916. The distribution of the attractiveness scores shows a slight positive skew (skew = 0.207) and a platykurtic shape (kurtosis = −0.663). The Shapiro–Wilk test was conducted to assess normality, and the test statistic yielded a *p*-value of 0.972, with a corresponding *p*-value of 0.954. These results indicated that the attractiveness score distribution for the COPD group can be considered approximately normal.

The HF group exhibited a moderately positive perception of the visual appeal of the application, with a mean attractiveness score of 1.341 (SD = 1.170). The distribution of attractiveness scores was slightly negatively skewed (−0.658), indicating that a majority of participants rated the attractiveness of the application higher than the mean. Furthermore, the kurtosis value of −0.455 suggested a relatively flat distribution compared to a normal distribution. The Shapiro–Wilk test (*p* < 0.001) indicated a deviation from normality, implying that the data distribution significantly deviates from a normal pattern.

Moving on to UEQ Perspicuity (Figure 9), the mean score for the COPD group was 1.464, with a median of 1.250 and a standard deviation of 0.982. The distribution of the perspicuity scores appeared to be relatively symmetric, as indicated by a skewness value of 0.200. The kurtosis value was −1.220, suggesting a distribution with negative kurtosis. The Shapiro–Wilk test yielded a test statistic of 0.972, with a corresponding *p*-value of 0.929, suggesting that the distribution of perspicuity scores was approximately normal. Participants in the HF group reported a relatively high level of perceived clarity and comprehensibility in the design of the application, as indicated by a mean perspicuity score of 2.063 (SD = 1.150). The distribution was negatively skewed (−1.243), suggesting that most of the participants perceived the application’s perspicuity positively. Kurtosis of 0.564 indicated a slightly peaked distribution compared to a normal distribution. The Shapiro–Wilk test (*p* = 0.001) suggested a departure from normality.

For the COPD group, the mean UEQ efficiency score was 0.881, with a median of 1.000 and a standard deviation of 1.147 (Figure 9). The efficiency score distribution showed a slight negative skew (skew = −0.312) and a kurtosis value of −0.364, suggesting a distribution that was relatively close to a normal shape. The Shapiro–Wilk test yielded a test statistic of 0.972, with a corresponding *p*-value of 0.978, which indicated that the distribution of efficiency scores was approximately normal.

The mean efficiency score of the HF group was 1.375 (SD = 1.151), which reflects a favourable perception of the efficiency of the application at the completion of the task. The distribution had a slightly negatively skewed shape (−0.383), implying that most participants perceived the application as efficient. A kurtosis of −0.863 indicated a distribution with flatter tails than a normal distribution. The Shapiro–Wilk test (*p* = 0.381) indicated a departure from normality.

The COPD group had a mean UEQ dependability score of 0.988, with a median of 0.500 and a standard deviation of 1.198 (Figure 9). The distribution of the dependability scores appeared slightly positively skewed (skew = 0.298) and exhibited a kurtosis value of −1.081, suggesting a platykurtic distribution. The Shapiro–Wilk test yielded a test statistic of 0.972, with a corresponding *p*-value of 0.936, indicating that the distribution of dependability scores can be considered approximately normal.

Participants in the HF group reported positive perceptions of the reliability of the application, with a mean dependability score of 1.700 (SD = 1.015). The distribution showed a slightly negatively skewed shape (−0.301), indicating that most of the participants perceived the application as reliable. The kurtosis of −1.022 suggested a relatively flat distribution with thin tails. The Shapiro–Wilk test (*p* = 0.160) suggested a departure from normality.

For the COPD group, the mean UEQ stimulation score was 0.821, with a median of 0.750 and a standard deviation of 0.949 (Figure 9). The distribution of stimulation scores showed a slight negative skew (skew = −0.305) and a kurtosis value of 0.311, indicating a distribution that was relatively close to normal. The Shapiro–Wilk test yielded a test statistic of 0.972, with a corresponding *p*-value of 0.981, suggesting that the distribution of the stimulation scores was approximately normal.

The mean stimulation score in the HF group was 1.125 (SD = 1.111), suggesting that a moderate level of perceived stimulation was provided by the characteristics of the application. The distribution was relatively symmetrical, as indicated by a skew of −0.086. A kurtosis of −1.387 implied a flatter distribution than a normal distribution with thin tails. The Shapiro–Wilk test (*p* = 0.104) suggested a departure from normality.

The COPD group had a mean UEQ novelty score of 0.929, with a median of 1.000 and a standard deviation of 0.867 (Figure 9). The distribution of novelty scores exhibited a negative skew (skew = −1.159) and a kurtosis value of 1.916, suggesting a distribution with a long left tail and a leptokurtic shape. The Shapiro–Wilk test yielded a test statistic of 0.972, with a corresponding *p*-value of 0.921, indicating that the distribution of novelty scores could be considered approximately normal.

The HF group reported a mean novelty score of 0.775 (SD = 1.029), indicating a moderate perception of the novelty of the application. The distribution was relatively symmetric, with a skew of −0.145. A kurtosis of 0.280 suggested a slightly peaked distribution compared to a normal distribution. The Shapiro–Wilk test (*p* = 0.607) suggested a departure from normality.

For the COPD group, the mean pragmatic quality score was 0.476, with a median of 0.500 and a standard deviation of 0.774 (Figure 9). The distribution of pragmatic quality scores shows a negative skew (skew = −0.840) and a kurtosis value of 0.746, indicating that the distribution was slightly skewed to the left and exhibited a relatively flat shape. The Shapiro–Wilk test yielded a test statistic of 0.972, with a corresponding *p*-value of 0.937, suggesting that the distribution of pragmatic quality scores was approximately normal.

The mean pragmatic quality score of the participants in the HF group was 0.662 (SD = 0.660), reflecting a favourable perception of the practical usefulness of the application. The distribution was relatively symmetrical, with a skew of −0.149. A kurtosis of −0.981 indicated a distribution with flatter tails than a normal distribution. The Shapiro–Wilk test (*p* = 0.079) suggested a departure from normality.

The COPD group had a mean UEQ hedonic quality score of 0.726, with a median of 0.750 and a standard deviation of 1.003 (Figure 9). The distribution of the hedonic quality scores exhibited a negative skew (skew = −0.560) and a kurtosis value of −0.082, indicating a distribution that was slightly skewed to the left and had a relatively flat shape. The Shapiro–Wilk test yielded a test statistic of 0.972, with a corresponding *p*-value of 0.951, suggesting that the distribution of hedonic quality scores could be considered approximately normal.

The mean hedonic quality score among the HF group was 1.063 (SD = 0.862), suggesting a positive perception of the application’s ability to provide enjoyable interactions. The distribution was slightly positively skewed (0.707), indicating that more participants positively rated the application’s hedonic quality. A kurtosis of −0.392 implied a distribution with flatter tails. The Shapiro–Wilk test (*p* = 0.069) suggested a departure from normality.

For the COPD group, the mean overall score was 1.096, with a median of 1.125 and a standard deviation of 0.887 (Figure 10). The distribution of the overall scores showed a positive skew (skew = 0.759) and a kurtosis value of −0.653, indicating a slight skew to the right and a platykurtic shape. The Shapiro–Wilk test yielded a test statistic of 0.972, with a corresponding *p*-value of 0.925, suggesting that the distribution of overall scores was approximately normal.

The average overall score in the HF group was 0.864, indicating a relatively high level of overall user experience satisfaction. The median score was 0.940, and the standard deviation was 0.652, suggesting some variability in the participant ratings. A skew value of 0.150 suggested a slightly positive skew, while a kurtosis value of −0.445 indicated a slightly platykurtic distribution. The Shapiro–Wilk test was not significant (*p* = 0.787), indicating that the data were normally distributed.

The COPD group had a mean change in necessity score of 0.333 (SD = 2.526), indicating a minor change in the perceived necessity of the application features (Figure 10). The distribution was moderately positively skewed (0.885), suggesting that most of the participants experienced a slight increase in perceived necessity. A kurtosis of 2.979 indicated a distribution with relatively heavy tails compared to a normal distribution. The Shapiro–Wilk test (*p* = 0.090) suggested a departure from normality.

In the HF group, the mean change in necessity score was 0.105 (SD = 2.622), also indicating a minor shift in perceived necessity. The distribution was relatively symmetrical, as indicated by a skew of 0.513. A kurtosis of 0.078 suggested a flatter distribution than a normal distribution. The Shapiro–Wilk test (*p* = 0.543) indicated no significant difference from normality.

The COPD group reported a mean concern change score of −1.133 (SD = 2.774), suggesting a decrease in perceived concerns about the use of the application (Figure 10). The distribution was moderately negatively skewed (0.043), indicating that most of the participants experienced a reduction in concerns. A kurtosis of 1.253 indicated a distribution with heavier tails. The Shapiro–Wilk test (*p* = 0.080) suggested a departure from normality.

The mean concern change score for the HF group was −0.763 (SD = 2.312), indicating a similar reduction in perceived concerns. The distribution was negatively skewed (0.388), indicating that most of the participants experienced decreased concerns. A kurtosis of −0.384 suggested a distribution with flatter tails. The Shapiro–Wilk test (*p* = 0.415) indicated no significant difference from normality.

The COPD group reported a mean NCD (Necessity–Concerns Differential) change score of 0.257 (SD = 0.776), suggesting a minor shift in perceived NCD of BMQ (Figure 10). The distribution was moderately positively skewed (1.387), implying that most of the participants perceived a slight increase in NCD. A kurtosis of 3.427 indicated a distribution with relatively heavy tails. The Shapiro–Wilk test (*p* = 0.097) suggested a departure from normality.

In the HF group, the mean NCD change score was −0.091 (SD = 1.154), indicating a minor decrease in the perceived NCD change. The distribution was strongly negatively skewed (−2.410), suggesting that most of the participants experienced a reduction in perceived change in NCD. A kurtosis of 8.744 indicated a distribution with much heavier tails. The Shapiro–Wilk test (*p* < 0.001) indicated a significant departure from normality.

The COPD group reported a mean SUS score of 68.333 (SD = 13.401), indicating a generally favourable perception of the usability of the Medimonitor application. The distribution was slightly positively skewed (0.207), suggesting that most of the participants rated the usability of the application favourably (Figure 11). A kurtosis of −0.889 indicated a distribution with relatively lighter tails than a normal distribution, implying a moderate degree of peakness. The Shapiro–Wilk test (*p* = 0.562) indicated no significant difference from normality.

In the HF group, the mean SUS score was 76.875 (SD = 18.671), indicating a higher mean usability perception than in the COPD group. The distribution was moderately negatively skewed (−1.383), suggesting that most participants in this group also favourably rated the application’s usability. A kurtosis of 2.352 indicated a distribution with relatively heavier tails, implying a flatter peak. The Shapiro–Wilk test (*p* = 0.013) suggested a departure from normality.

The results of the UEQ benchmark for HF patients indicated a pragmatic quality score of 0.663, categorising it as “bad” and falling within the range of the 25% worst results. In terms of hedonic quality, HF patients had a score of 1.063, indicating that they were above average (Figure 12). This finding suggests that 25% of the results were better, while 50% were worse. The overall user experience for HF patients received a score of 0.86, categorising it as below average, with 50% of the results being better and 25% being worse.

In contrast, the COPD UEQ benchmark revealed a pragmatic quality score of 0.476, placing it in the “bad” category and within the range of the 25% worst results. The hedonic quality score for COPD patients was 0.726, indicating that it was below average. This implies that 50% of the results were better, while 25% were worse. The overall user experience for COPD yielded a score of 0.60, which was also categorised as below average, with 50% of the results being better and 25% being worse.

In general, the skewness and kurtosis values for the aforementioned usability variables in both groups showed varying degrees of departure from the normal distribution. The HF group had more extreme skewness and kurtosis values for the SUS score, indicating a potential non-normal distribution in usability perceptions among HF patients. On the other hand, the UEQ showed skewness and kurtosis values closer to normal in both groups.

When comparing the UEQ benchmark results between HF and COPD patients, it is evident that both conditions have suboptimal pragmatic quality scores, indicating a less favourable user experience in terms of meeting functional and practical needs. However, it should be noted that HF patients scored higher in terms of pragmatic quality (0.663) than the COPD patients did (0.476), suggesting that HF patients perceive a relatively better fulfilment of their pragmatic requirements.

In terms of hedonic quality, HF patients had a greater score (1.063) than the COPD patients did (0.726), indicating a more positive subjective experience for patients with HF. This could be attributed to various factors such as the nature of the condition, available treatment options, or differences in the management of symptoms. However, further investigation is necessary to understand the specific factors that contribute to these contrasting results.

The overall user experience scores for both conditions indicated that there is room for improvement. Although HF patients had a slightly higher overall score (0.86) than the COPD patients did (0.60), both of these scores fell within the below-average category. This suggested that efforts should be made to enhance the overall user experience for individuals affected by these conditions.

For the COPD group, the mean MARS change score was 0.591, with a median of 0.000 and a standard deviation of 1.532 (Figure 13). The distribution of MARS change scores showed a positive skew (skew = 0.073) and a kurtosis value of 1.337, indicating a distribution that was slightly right-skewed and had a leptokurtic shape. The Shapiro–Wilk test yielded a test statistic of 0.953, with a corresponding *p*-value of 0.888, suggesting that the distribution of MARS change scores can be considered approximately normal.

The pilot study aimed to explore the correlations between user experience, measured through the SUS and UEQ, and patient adherence, assessed by changes in MARS scores. Furthermore, the study included an examination of the perceived necessity of and concerns about specific prescribed medications, as measured by the BMQ at the beginning and end of the pilot study. The results, based on a sample size of 41 participants, were as follows:

When examining the relationship between SUS scores and adherence, the correlation was found to be negative but weak. The Spearman’s rho was −0.144, However, not reaching statistical significance (*p* > 0.05). This finding suggests that there is no strong association between SUS scores and changes in adherence.

Similarly, the correlations between the appraisal of pragmatic quality (measured by the UEQ) and adherence were weak and not statistically significant and with a Spearman’s rho equal to −0.118.

The appraisal of hedonic quality, as measured by the UEQ, also showed weak and non-significant correlation with adherence as described by a Spearman’s rho equal to −0.143.

There is no significant correlation between the changes in patients’ perception of necessity and the changes in adherence described by Spearman’s rho equal to −0.113. This finding indicates that the changes in patients’ perception of the necessity of the prescribed medication did not significantly affect adherence.

Similarly, the correlation between changes in concerns about prescribed medications and adherence was negative but weak with Spearman’s rho equal to −0.087. However, this correlation is also not significant.

In summary, the results of this pilot study did not indicate any strong associations between user experience (measured by the SUS and UEQ) and patient adherence (measured by changes in MARS scores) (Table 3). Furthermore, changes in patients’ perception of the necessity of and concerns about prescribed medication, as measured by the BMQ, did not demonstrate significantly influence adherence. These findings suggest that factors other than user experience, perceived necessity, and concerns may play a more influential role in determining patient adherence to treatment regimens. Further research with larger sample sizes and additional variables is necessary to gain deeper insights into the complex relationships among user experience, beliefs about medication, and adherence in the healthcare context.

Linear regression analysis was aimed at investigating the relationship between various factors and patient adherence, as measured by changes in MARS scores. 

First, the model fit measures indicated that the model explained a moderate amount of variance in the data (Table 4). The coefficient of determination (R-squared) was 0.451, suggesting that approximately 45.1% of the variance in the MARS change could be explained by the independent variables. An adjusted R-squared of 0.324 accounted for the number of predictors in the model and penalised overfitting.

The *p*-values of the individual predictors revealed that none of the predictors had a statistically significant effect on the MARS change (Table 5). The SUS score, which represents user experience, had a *p*-value of 0.244, indicating that it was not a significant predictor. Similarly, changes in perceived necessity of and concerns about the medication were not significant predictors (*p* = 0.564 and *p* = 0.178, respectively). However, the MARS baseline score was found to be a significant predictor with a *p*-value of less than 0.001.

The following assumptions were made (see Table 6 and Table 7):Normality: Normality tests (Shapiro–Wilk) indicated that the residuals did not strictly follow a normal distribution). This suggests a deviation from the assumption of normality, which may affect the validity of the regression estimates.Heteroskedasticity: The heteroskedasticity tests (Breusch–Pagan) did not provide evidence of significant heteroskedasticity in the residuals. This suggests that the assumption of homoscedasticity is reasonable.Autocorrelation: The Durbin–Watson test showed a value of −0.258, indicating the possibility of autocorrelation in the residuals. However, the *p*-value of 0.160 suggests that the presence of autocorrelation is not statistically significant.Collinearity: Collinearity statistics (Variance Inflation Factor and Tolerance) indicated no severe multicollinearity among predictor variables. All the variables had VIF values of less than two and tolerance values greater than 0.5, suggesting that collinearity is not a major concern.

In summary, the linear regression analysis provided limited support for the relationship between the factors examined and patient adherence. While the model explained a moderate amount of variance, none of the individual predictors were found to be significant in predicting changes in adherence. Furthermore, assumptions related to normality, autocorrelation, and collinearity showed deviations from the ideal conditions, which may affect the reliability of the regression estimates.

Another objective of this study was to compare user experience (UX) and adherence between two groups of patients, specifically those with chronic heart failure (HF) and patients with chronic obstructive pulmonary disease (COPD). The results of the independent samples’ *t*-test reveal several interesting findings (see Table 8).

In terms of UX, the following three dimensions were assessed using the User Experience Questionnaire (UEQ): Hedonic Quality, Pragmatic Quality, and Overall UX. The *t*-test results reveal no significant differences between the HF and COPD groups in Hedonic Quality (*t* = −1.149, *p* = 0.258), Pragmatic Quality (*t* = −0.827, *p* = 0.413) or Overall UX (*t* = −1.229, *p* = 0.226). These findings suggest that both groups perceive a similar level of UX when interacting with the system. 

To evaluate adherence, the MARS was used to measure baseline adherence, adherence at the end of the pilot, and change in adherence. The *t*-test results show no significant differences between the HF and COPD groups in terms of baseline adherence (*t* = −0.532, *p* = 0.598), adherence at the end of the pilot (*t* = 0.663, *p* = 0.511) or change in adherence (*t* = 1.136, *p* = 0.262). These findings suggest that both groups exhibit similar levels of self-reported adherence to their medication regimen. 

Additionally, the *t*-test results for Necessity Change (*t* = 0.256, *p* = 0.800) and Concern Change (*t* = −0.425, *p* = 0.674) indicate no significant differences between the HF and COPD groups. These findings suggest that both groups experienced similar changes in their perceptions of medication necessity and concern. 

Furthermore, the *t*-test results for NCD Change (*t* = 1.001, *p* = 0.324) reveal no significant differences between the HF and COPD groups in terms of the change in the necessity–concerns differential of BMQ. This suggests that both groups report similar perceptions of beliefs about medication.

Finally, the *t*-test results for the SUS score (*t* = −1.689, *p* = 0.099) indicate no significant differences in the perceived usability of the system between the HF and COPD groups. Both groups report similar levels of usability. 

In summary, the results of the *t*-test analysis comparing UX and adherence between the HF and COPD groups suggest that there are no significant differences between the two groups. Both groups had reported similar perceptions of UX, adherence, medication necessity and concerns, change in negative consequences, and system usability. These findings indicate that the system was equally well-received and effective in both patient populations, highlighting its potential to support adherence to medications in contexts of different chronic diseases.

We used jamovi for statistical analysis [50,51,52,53].

### 3.2. Post Pilot Follow-Up Qualitative Research with Pilot Participants

Based on the transcribed remarks, we identified recurring themes and gained insight into aspects related to the overall evaluation of the study, user experience, perceived benefits, adaptation time, and technical issues. The topics are discussed below, specific to each of the groups of patients. The numbers in brackets indicate how many people chose that answer.

#### 3.2.1. HF Use Case

According to the overall evaluation of the study, nine patients in this group expressed satisfaction and two patients were unsatisfied. Patients mainly highlighted the smooth functioning and low workload of the measurement process. The reasons for the dissatisfaction were lack of comfort due to technical issues, sense of duty, difficulty combining work and measurement, or questionnaires repeated too frequently.

The patients (nine) also highlighted several benefits they perceived from participating in the study. Patients mentioned receiving up-to-date health data from regular measurements of their weight (five), blood pressure (one), and oxygen saturation (one), which provided them with an overview of their health status. Other perceived benefits included a regular regime (two) and motivation to walk daily (one). However, some patients (two) mentioned that they did not understand the data obtained from the ECG and could not interpret them (two). Furthermore, for some patients, the purpose of the study was to provide researchers with values about their health status, but they did not perceive any benefit (two).

The perceived user experience varied among patients, from “it was truly good to work with” to “it was not very comfortable, and it took quite a long time”. The time required for daily measurements also varied, from approximately 15 min (two) to one or two hours (two) in some cases. The patients also mentioned the additional time required to learn to use the devices and application. “After a while, I found out that I cannot measure myself in the morning, that I have to measure myself in the evening” (because of the daily steps count).

An important topic related to user experience was technical issues. A total of nine patients reported experiencing technical problems, including non-functioning, unpaired, or discharging weight scale (four), disconnected smartwatches (two), not transmitting ECG (two), or overall problems with logging to the application or data uploading (three).

#### 3.2.2. COPD Use Case

The patients’ experiences with the Medimonitor application used for COPD monitoring were generally positive. Most of the patients (nine) expressed satisfaction with the process, stating that they were glad to participate. They mentioned activation, daily routine, and “instrumental support in the required wide range and quality”. Some of the participants (two) mentioned problems with the functionality of the application that made them feel uncomfortable. For some, the questionnaires were repeated too often (two), and the general use of the application “was too complicated at certain moments” (one).

The perceived benefits of participating in the study included “more frequent measurements” (one), obtaining a “blood pressure overview” (one), remote evaluation of measurements (one), the ability to order medication through the Medimonitor application (one), and refining or adjusting the diagnosis by the physician based on the health data obtained (two). However, similar to the previous group, some patients (two) reported that they did not understand the measured data (e.g., spirometry) and were unable to interpret them. One patient did not perceive any benefits from the measurements conducted in the study and stated: “I already sleep with the breathing device due to COPD, so I can view my sleep data in the mobile phone application“.

The overall user experience was rated lower than that in the HF group. Only two participants expressed satisfaction with the user-friendliness of the devices used. Five participants were partially satisfied due to issues with some devices but overall found it acceptable. Six participants were dissatisfied. They cited obstacles such as the need for assistance from a family member (two), manual data entry (two) (non-functional automatic transmission), login problems with the application (one), or repeated blood pressure measurements (one). Patients also mentioned that “it was too complicated at some points” (one) or expressed the opinion that “it can be a challenge for people over 65 years of age to use this technology” (two). One participant experienced vision problems that made it difficult to work with the tablet. The adaptation time varied, according to the participants, from two or three days to one week.

In the context of reduced user-friendliness, technical issues were a major topic. The most problematic complication was the use of a ring oximeter (nine). Patients reported issues such as non-functioning, fast discharge, and long or non-functional uploading. Five participants reported problems with the application. They were related to logging in, data uploading, or freezing of an application that had to be restarted. Updates were also considered problematic. The patients encountered some issues with data transmission via inhalation (two) or spirometry (one). However, technical support was positively viewed as fast working (one).

## 4. Discussion and Study Limitations

As mentioned in the introduction, previous studies suggest a link between the usability and user experience of eHealth applications and patient adherence to medication. Key issues such as usability, accessibility, reliability, versatility, and user experience require further attention from researchers and practitioners. These findings emphasise the importance of considering these factors in the development of eHealth applications to enhance medication adherence [1,2,3].

The findings reported in this work highlight the need for further research and the consideration of additional variables to gain a deeper understanding of the complex factors that influence patient adherence. This is also related to health and ICT literacy of the patients. Due to the fact that patients had to fill in various other questionnaires and the activities were synchronised across the European union, it was decided to optimise the number of questionnaires ultimately delivered to Medimonitor users. 

Other questionnaires administered to the patients focused, among others, on questions related to family, social, and economic status. Disease severity was also measured for both the COPD and heart failure patients. Their QoL (Quality of Life) was measured as well by EQ-5D and WHOQOL, at both baseline and after the pilot testing. Nevertheless, the main goal of the presented manuscript was whether the Medimonitor UX itself, independent of the patient’s status or QoL, can provide more with the increase in their adherence. Therefore, these data were not included in the current article.

The results of MARS-5 questionnaire show only a small improvement in adherence. The mean values of MARS-5 change between baseline and end-of-pilot for COPD and HF patients are 0.591 and 0.043, respectively. That indicates an increase in adherence among COPD patients; however, based on the *t*-test, the increase is statistically not significant. It needs to be stressed that the room for improvement is rather small due to the fact that the baseline adherence, as measured by MARS-5, for COPD and HF patients are 23.318 and 23.565, respectively. Values between 24 and 25 mean highly adherent patients. Though the exact cut-off scores might differ for various diseases. The high initial value of adherence makes the assessment of UX influence on adherence even more challenging. Also, as the results are based on the self-reported questionnaires, the patients might not be giving the researchers truthful responses. Considering the fact, that the electronically monitored adherence has not proved to be feasible with both cohorts, the low accuracy in self-reported adherence might be the true source of the non-significant results. The smart pillboxes were not used correctly, or were not used at all, as the patients were not satisfied with its design. And the smart inhalers were also not used correctly to provide with the usable data. It shows that even the usability design of the devices intended for the adherence measurement is crucial. At least the resulting data, though gathered through the self-reported questionnaires, did not show any decrease in the adherence. The researcher’s vigilance in similar cases is highly recommended to cover all possible sources of inaccuracies, which in the case of the adherence monitoring might be extremely challenging. Though more precise approaches might be developed, they need to be respectful towards the patients. Respecting the patient’s privacy and free will, even if they do not participate in their treatment regimen, is crucial.

Another important aspect that is assessed in parallel with the adherence are beliefs about medicines measured by the BMQ. The most important parameter in the assessment is the comparison of the baseline and end-of-pilot Necessity–Concerns Differential, designated as the NCD Change. The NCD change’s mean values are 0.257 and −0.091 for COPD and HF, respectively. Though the improvement in NCD is observed for the COPD patients, the differences among the COPD and HF cohorts are not statistically significant. In general, it can be said that the necessity slightly increased and concerns slightly decreased for both groups, which is a positive effect of the digital solution provided to both groups. 

User experience was measured by the UEQ and SUS. An advantage of both tools is that there are available benchmarks. Nevertheless, they are not exclusively related to eHealth or mHealth solution. Hence, they can provide only a rule-of-thumb comparison against other available products or services. For UEQ, there are three distinctive groups, namely the general benchmark, business software, websites and web services. In the presented case, the general benchmark was applied. Unlike the BMQ, the UEQ results show that the perception of the digital solution was better among the HF cohort. The mean values for the pragmatic and hedonic qualities and overall are 0.66, 1.06, and 0.86, respectively. An above average result is acquired only in the hedonic quality scale. The pragmatic quality is bad in comparison to benchmark. The overall results for the HF cohort are below average. Though an improvement in NCD was observed in the COPD cohort, their perception of UX, as measured by the UEQ, is even worse than those of the HF cohort. The COPD cohort’s results for UEQ in terms of the pragmatic and hedonic qualities and overall are 0.48, 0.73, and 0.6, respectively. The pragmatic quality is observed to be bad and the hedonic quality as well as overall appraisal are below average. In that matter, it seems that the NCD change and UX, as measured by the UEQ, are detached. 

Similar to the UEQ, the resulting mean values of SUS are 68.33 for COPD cohort and 76.88 for HF cohort, giving C and B grades, respectively. For an excellent design, the result should exceed value of 80.8. The cut-off score for an acceptable solution is set above 70.0 and is unacceptable below 50.0. Hence, the presented solution is only marginally acceptable as perceived by the COPD cohort and acceptable for the HF cohort, again giving a result that does not correspond to those acquired by the MARS or BMQ. It needs to be stressed that usability is not an absolute construct and should be understood in the broader context of its particular field of application. The requirements placed on the mHealth applications are, of course, high as they are about to be used for medical purposes, and typically by patients that might be suffering from some other limitations that may not allow them to use the application in the same way as the healthy population might. In that sense, it is important to reflect on standards like ISO 9241-11 that define the usability. It is impossible to define the system’s usability if the prospective users and their characteristics, including the characteristics of their environment, are not perfectly understood and described to developers. The approach outlined in the ISO standards was also used during the Medimonitor development. 

If the user experience is the only variable, based on the metrics applied in this study, the HF cohort should achieve better adherence improvement over the COPD cohort. In contradiction to that expectation, the COPD cohort achieved better results in terms of self-reported adherence and beliefs towards medication, even though they did not perceive the digital solution as well as the HF cohort did.

The only reliable predictor in the linear regression model to infer from regarding the change in adherence measured by MARS-5 is a baseline MARS-5 value. The rest of the predictors have no statistical significance. 

Based on the aforementioned results, it can be seen that other variables are influential in the patients’ adherence as well.

The main limitation of the present study was the relatively small number of participants involved, which were further limited by the unavailability of resources and the relatively short duration of the study. Moreover, as the pre-pilot and pilot activities occurred during the COVID pandemic and the project timeframe was already extended due to this fact, it was not possible to add more participants. The project spanned over 3 months of testing with the patients monitoring themselves by various medical devices as capturing the health-related data from the portable medical devices was a main goal of this pilot study, while the UX was an integral part of the pilot to achieve the best patient engagement in their regimen. Though the number of participants was limited, the long-term nature of the mHealth application testing still makes the study rather unique and the chosen approach unusually holistic.

Additional study limitations included difficulty in isolating each aspect of the study. Even though appropriate measures, such as validated questionnaires, were applied, it was often difficult to study each aspect of the study in isolation. Therefore, the average and below average results acquired by the SUS and UEQ questionnaires may be caused by service issues (e.g., connectivity issues, application stability) rather than by the handling of the application. It might be possible that the patients’ opinions are modulated by the healthcare service quality in general, or even by the personality of the administering personnel. In the future, it might be useful to apply methods to assess the personality of patients as well as the administering physicians. The degree of expected patient involvement should be assessed in the context of their personality, as well as any observed discrepancies between the levels of reasonable application and device usage, and in the frequency of the expected feedback. Participants with higher expected physician’s or nurse’s feedback were usually open to higher level of participation. Throughout the study, a mixed methods design approach was employed. In the beginning of the Medimonitor UX research and development, the qualitative methods were used in a rather exploratory manner to establish the basic framework for further development of the Medimonitor. During and after the several optimisation loops, the application was delivered for pilot use by the chronically ill patients. During and immediately after the pilot, mostly quantitative methods were applied. After the pilot testing, debriefing was performed using a qualitative approach once again to gather explanatory insight into the quantitative results.

## 5. Conclusions

This article describes the results of a multi-stage, mixed methods approach to assess the influence of UX on adherence to pharmacotherapy among chronically ill patients. The applied methodology and results may assist mHealth application developers, as well as clinical experts, who can design an efficient technical solution that is well suited to the needs of particular types of patients based on their disease status, age, gender, and other specifics.

The analysis of the HF and COPD groups revealed no significant differences in terms of UX, adherence, or perceived usability. This indicated that the system was equally effective in supporting medication adherence for both patient populations.

Furthermore, the variables measured in this study, such as hedonic and pragmatic qualities, necessity and concern change, and negative consequences of medication, were not found to be strong predictors of adherence, as measured by the self-reported questionnaires in the context of HF and COPD. Other factors not considered in this study may have had a more significant impact on adherence.

The HF and COPD groups reported similar changes in medication concerns, negative consequences, and overall user experience when using the system. This suggested that the system had a comparable effect on these aspects for both patient populations.

According to the qualitative evaluation after the pilot study, there was mixed satisfaction among patients using the Medimonitor application. Most patients appreciated the smooth functioning and benefits, such as receiving up-to-date health data and maintaining a regular health regime. However, dissatisfaction arose from the technical issues, lack of comfort, and difficulties balancing work and measurement tasks. Usability varied, with some patients finding the devices user-friendly, while others faced significant challenges and technical problems, such as issues with weight scales, smartwatches, and data uploading. In summary, the findings suggest that the system examined demonstrated similar effectiveness and usability in supporting medication adherence in patients with HF and COPD. However, the variables examined in this study may not be strong predictors of adherence. Further research is needed to explore additional factors that could influence adherence behaviours in these patient populations.

For further studies, working with larger groups for an extended period of time is recommended. Furthermore, the use of technology for the measurement of adherence might be more effective than relying solely on self-reported questionnaires. In this specific case, the application of technology was not successful, as the users simply did not use them at a sufficient magnitude to generate a sufficient amount of data for them to be statistically assessed. The following question remains: If the participants recognise that their adherence is already being measured, do they adjust their adherence, hence making the technology redundant in terms of measurement? Is it necessary to undertake the work in a blinded manner to obtain robust results?

## Figures and Tables

**Figure 1 behavsci-14-00489-f001:**
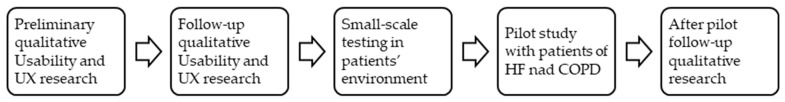
The development and testing sequence of the application.

**Figure 2 behavsci-14-00489-f002:**
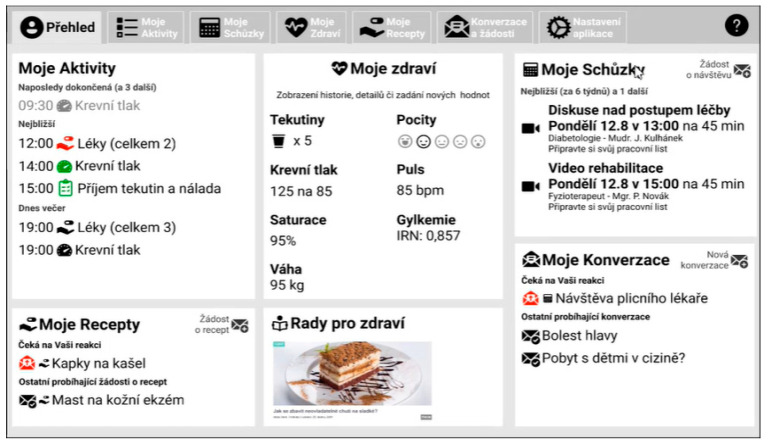
First version of the wireframe prototype (from top left: My activities—list of completed and scheduled activities, including medication intake and blood pressure measurements, My health—display of health metrics such as blood pressure, pulse, saturation etc., My appointments—details of upcoming medical appointments, from bottom left: My prescriptions—list of prescribed medications, Health tips—general health advice and tips, My conversations—communication with healthcare providers).

**Figure 3 behavsci-14-00489-f003:**
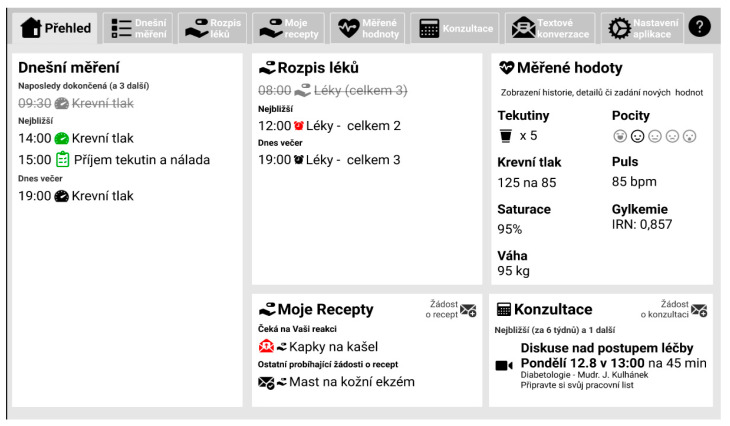
Second version of the wireframe prototype based on user feedback (from top left: Today’s measurements—list of completed and scheduled measurements, Medication schedule—details of scheduled medication intake for the day, My health metrics—display of health metrics, from bottom left: My prescriptions, Consultations—details of upcoming medical consultations).

**Figure 5 behavsci-14-00489-f005:**
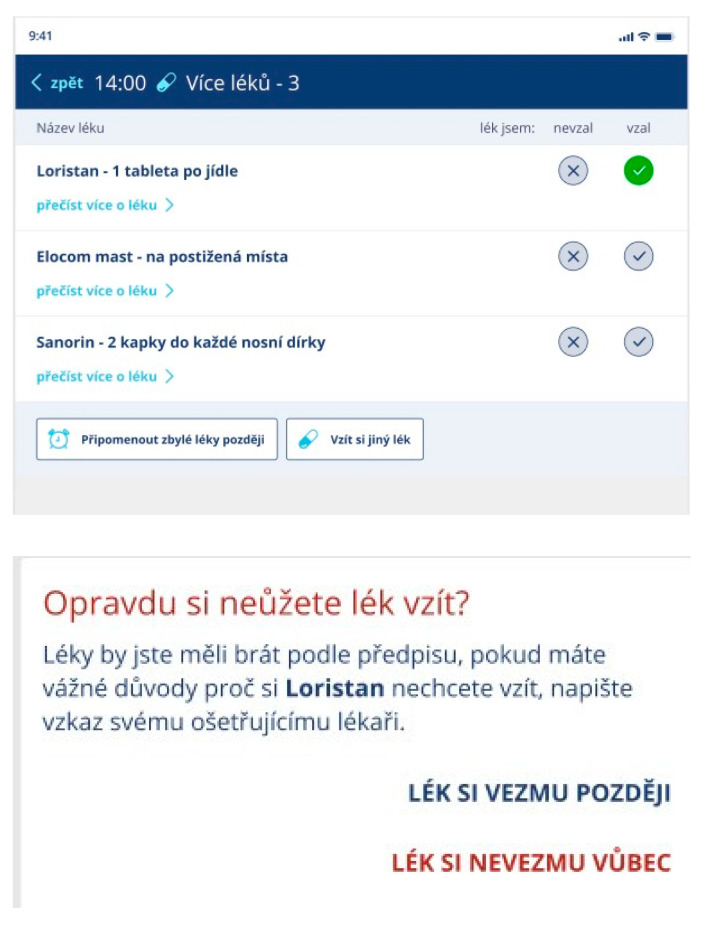
Confirmation of the medication consumption screen (on the top: confirmation of medication consumption, at the bottom: reminder about not taking medication).

**Figure 7 behavsci-14-00489-f007:**
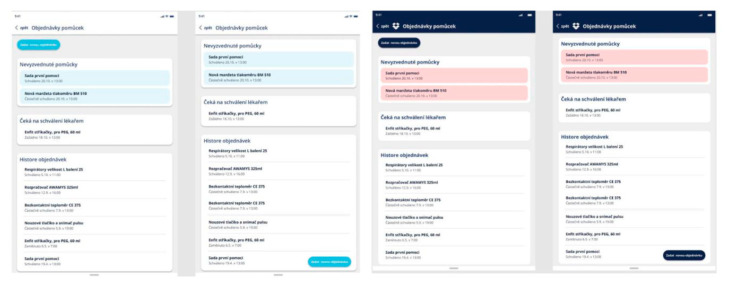
Medical aids ordering bright- (**left**) and dark-coloured screen variations (**right**).

**Figure 8 behavsci-14-00489-f008:**
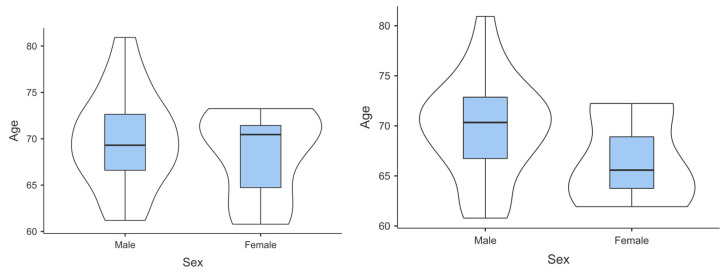
Age distributions of HF (**left**) and COPD (**right**) use case participants.

**Figure 9 behavsci-14-00489-f009:**
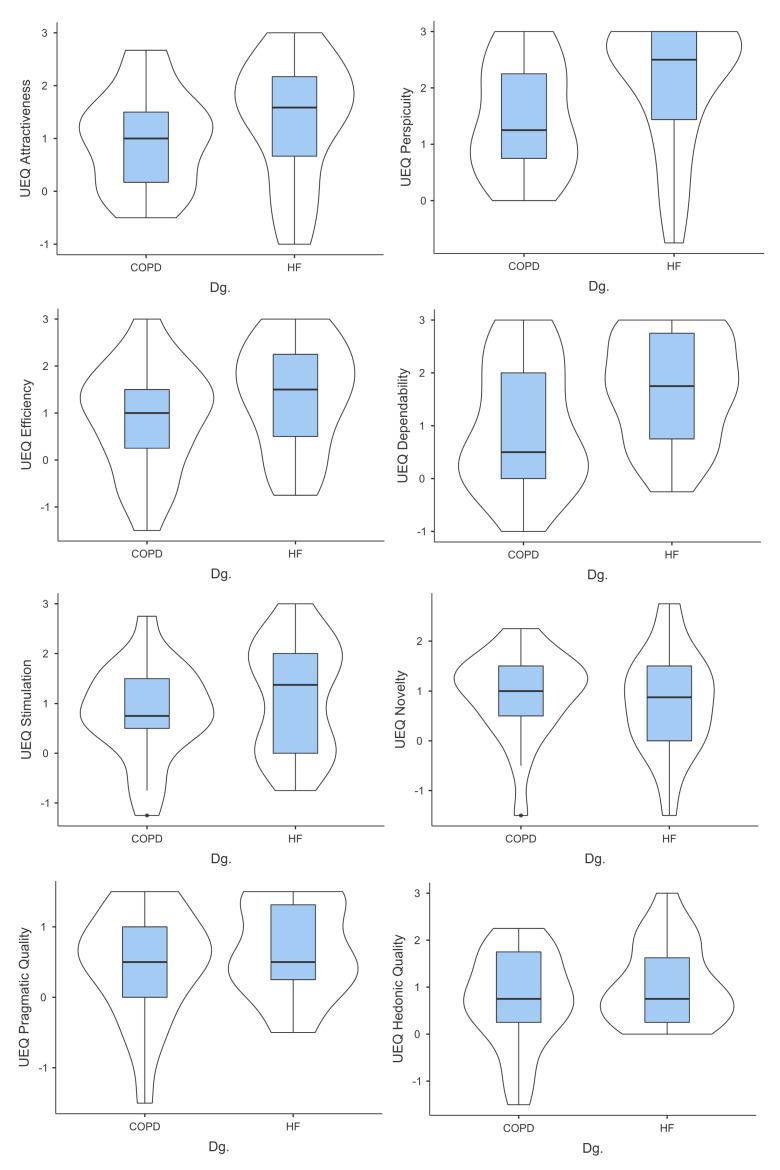
UEQ Dimensions (from **top left**: Attractiveness, **top right:** Perspicuity, **middle left:** Efficiency, **middle right:** Dependability, **lower left:** Stimulation, **lower right:** Novelty) and Quality (**bottom left**: Pragmatic, **bottom right**: Hedonic).

**Figure 10 behavsci-14-00489-f010:**
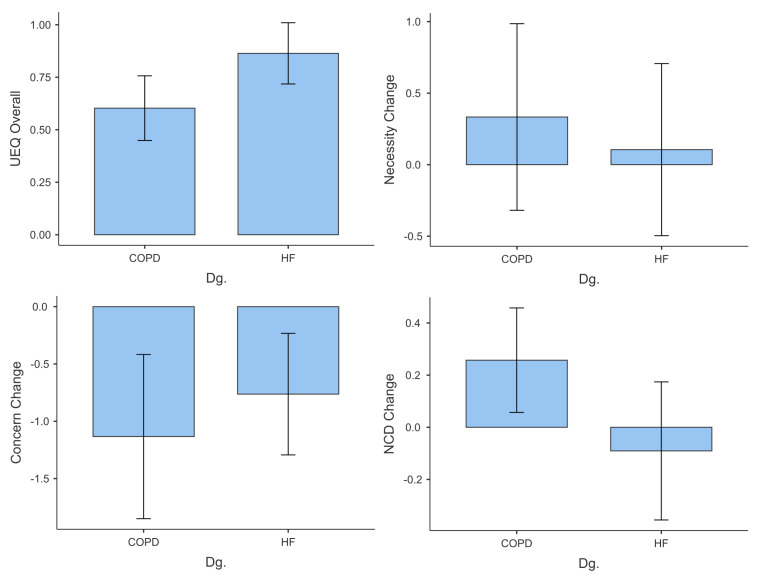
From top left: UEQ Overall score, BMQ Necessity score change, BMQ concern score change and BMQ NCD change for patients with COPD (**left**) and HF (**right**).

**Figure 11 behavsci-14-00489-f011:**
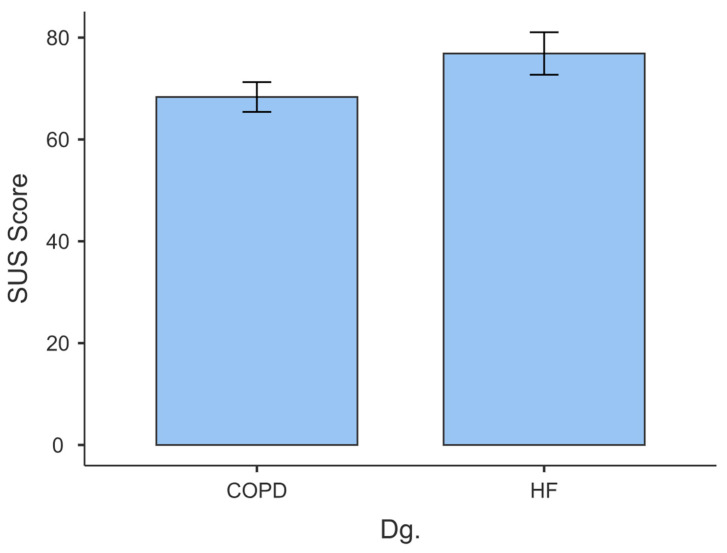
SUS score for patients with COPD (**left**) and HF (**right**).

**Figure 12 behavsci-14-00489-f012:**
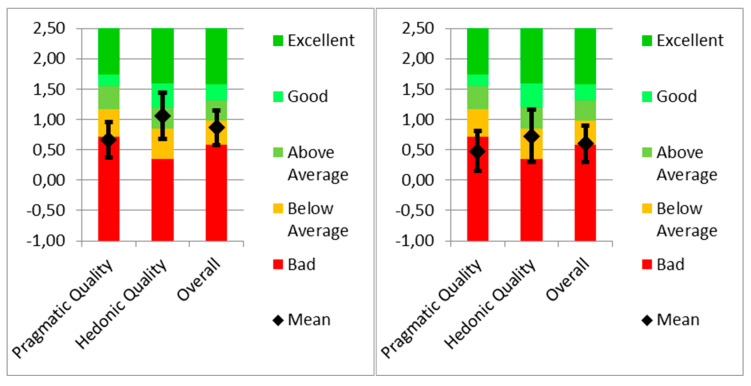
UEQ Short benchmark comparison of COPD (**left**) and HF (**right**) use cases.

**Figure 13 behavsci-14-00489-f013:**
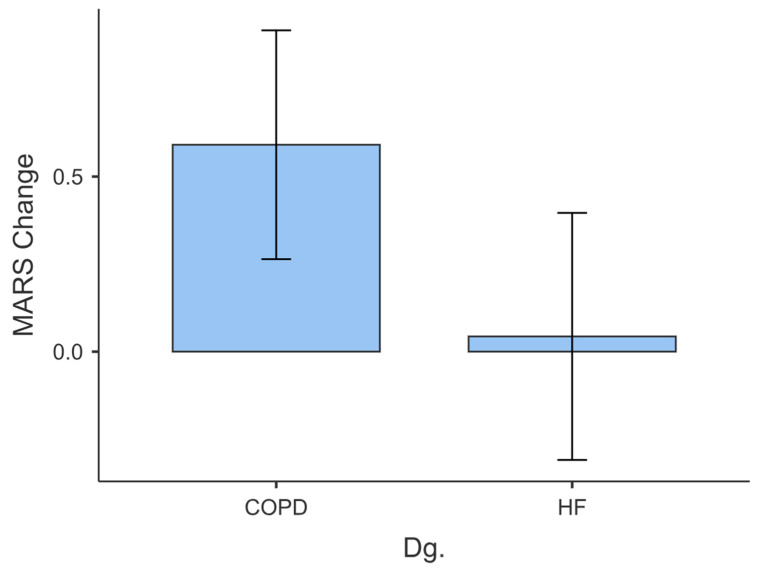
Change in the MARS score for COPD (**left**) and HF (**right**) patients.

**Table 1 behavsci-14-00489-t001:** Inclusion and exclusion criteria for HF use case.

Inclusion Criteria	Exclusion Criteria
LVEF ≤ 50%	LVEF ≥ 50%
Exclusion of obstructive coronary artery disease	NTproBNP ≤ 125 pg/mL
NTproBNP ≥ 125 pg/mL	Severe psychological disturbances
≥60 years of age	Absence of collaboration (informed consent)
Informed written and verbal consent	BP ≤ 110 mmHg without hypertensive medication

**Table 2 behavsci-14-00489-t002:** Inclusion and exclusion criteria for COPD use case.

Inclusion Criteria	Exclusion Criteria
COPD	Active smoking
≥60 years of age	Respiratory failure that requires oxygen therapy or ventilation support
Informed written and verbal consent	Severe psychological disturbances
Ability to participate in study activities	Absence of collaboration (informed consent)
	Other comorbid pulmonary disease
	Symptomatic heart failure
	Motor neuron diseases (e.g., amyotrophic lateral sclerosis etc.)

**Table 3 behavsci-14-00489-t003:** MARS, SUS, and UEQ correlation matrix.

		SUS Score	UEQ Pragmatic Quality	UEQ Hedonic Quality	Necessity Change	Concern Change
MARS Change						
	Spearman’s rho	−0.144	−0.118	−0.143	−0.113	−0.087
	*p*-value	0.370	0.461	0.373	0.523	0.625
	N	41	41	41	34	34

**Table 4 behavsci-14-00489-t004:** Linear regression—Model Fit Measures.

							Overall Model Test			
Model	R	R^2^	Adjusted R^2^	AIC	BIC	RMSE	F	df1	df2	*p*
1	0.671	0.451	0.324	121.092	133.064	1.189	3.555	6	26	0.011

**Table 5 behavsci-14-00489-t005:** Linear Regression—Model Coefficients for dependent variable MARS Change.

			95% Confidence Interval			
Predictor	Estimate	SE	Lower	Upper	*t*	*p*
Intercept	16.317	3.859	8.385	24.250	4.228	<0.001
SUS Score	−0.019	0.016	−0.053	0.014	1.192	0.244
Necessity Change	0.058	0.099	−0.146	0.262	0.585	0.564
Concern Change	−0.132	0.096	−0.329	0.064	−1.383	0.178
MARS Baseline	−0.623	0.157	−0.946	−0.300	−3.963	< 0.001
UEQ Pragmatic Quality	−0.063	0.420	−0.926	0.801	−0.149	0.883
UEQ Hedonic Quality	−0.116	0.286	−0.705	0.472	−0.406	0.688

**Table 6 behavsci-14-00489-t006:** Assumption checks for linear regression: Normality test (Shapiro–Wilk), Heteroskedasticity Test (Breusch–Pagan) and Autocorrelation test (Durbin–Watson).

	Statistic	*p*
Shapiro–Wilk	0.932	0.041
Breusch–Pagan	7.031	0.318
Durbin-Watson (Autocorrelation = −0.258)	2.505	0.160

**Table 7 behavsci-14-00489-t007:** Assumption check for linear regression: Collinearity Statistics.

	VIF	Tolerance
SUS Score	1.390	0.720
Necessity Change	1.125	0.889
Concern Change	1.025	0.976
MARS Baseline	1.048	0.954
UEQ Pragmatic Quality	1.291	0.775
UEQ Hedonic Quality	1.323	0.756

**Table 8 behavsci-14-00489-t008:** Independent Samples’ *t*-Test statistics.

		Statistic	df	*p*	Mean Difference	SE Difference
UEQ Hedonic Quality	Student’s *t*	−1.149	39.000	0.258	−0.336	0.293
UEQ Pragmatic Quality	Student’s *t*	−0.827	39.000	0.413	−0.186	0.225
UEQ Overall	Student’s *t*	−1.229	39.000	0.226	−0.261	0.212
MARS Baseline	Student’s *t*	−0.532	43.000	0.598	−0.247	0.465
MARS End of Pilot	Student’s *t*	0.663	43.000	0.511	0.300	0.453
MARS Change	Student’s *t*	1.136	43.000	0.262	0.547	0.482
Necessity Change	Student’s *t*	0.256	32.000	0.800	0.228	0.891
Concern Change	Student’s *t*	−0.425	32.000	0.674	−0.370	0.872
NCD Base	Student’s *t*	−0.525	32.000	0.604	−0.142	0.272
NCD End	Student’s *t*	0.646	32.000	0.523	0.204	0.316
NCD Change	Student’s *t*	1.001	32.000	0.324	0.348	0.347
SUS Score	Student’s *t*	−1.689	39.000	0.099	−8.542	5.057

## Data Availability

The data presented in this study are available upon request from the corresponding author. The data are not publicly available due to the formulations in the informed consent that the participants signed. The protocol of the study is available on the ClinicalTrials.gov website.

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
