# Peer review of "Patients’ UX Impact on Medication Adherence in Czech Pilot Study for Chronically Ill"

_behavsci, 2024, doi:10.3390/bs14060489_

Round 1

Reviewer 1 Report

Comments and Suggestions for Authors

This study investigated the relationship between baseline medication adherence, beliefs about medicines, user experience, and the impact of a disease-specific mHealth application on adherence changes in older adults with chronic heart failure (HF) and chronic obstructive pulmonary disease (COPD). The research employed a mixed methods approach, incorporating both quantitative and qualitative components. Despite expectations, the results did not demonstrate the anticipated relationship between the examined variables. However, qualitative findings revealed general patient satisfaction with the application, although technical issues during pilot testing may have influenced overall user experience perceptions. 

1. The small sample size of the article may limit the generalizability of the results and the power of statistical analysis. It is recommended to increase the sample size in the study.

2. Although the research results did not show the expected variable relationships, they provide valuable insights into the impact of user experience and technical issues on medication adherence. The authors are advised to further explore other factors that may affect adherence, such as patient education level, social support, and disease severity.

3. Article structure and clarity: It is recommended to condense the content, merge some figures and tables to highlight the main points of the article. The structure of the article should be clear for easy understanding by readers.

4. It is suggested that the authors provide more detailed analysis and explanations in the discussion section, as well as recommendations for future research directions.

Author Response

Dear reviewer,

thank you very much for reviewing our manuscript and giving us with insightful ideas for further improvement. We sincerely appreciate the feedback that you have given to us.

Based on your comments we have been able to substantially improve the manuscript.

On behalf of the co-authors, We would also like to respond to the specific remarks you have raised:

  1. We agree, that the size might influence the generalizability of the results. However, due to the nature of the project (there were multiple pilot sites across the European Union) it was not possible to recruit more participants, which would require either a higher budget (each participant was equipped with a tablet and several medical devices) or time (so we could rotate devices among participants). The latter one was also influenced by COVID-19, as the project was conducted during the global pandemic. Also, it is not possible to increase the size of the group now as the project has officially ended and there are unfortunately no remaining resources, either financial or personal, to conduct the research with more participants.
  2. We appreciate that as we also think that despite that the direct relationship was not found, the insights still provide useful experience for other researchers and developers. The participants‘ education level was only measured by the single-item questionnaire assessing what is confidence in filling out the health-related forms. Regarding social support, the Gijón scale was administered. The Gijón scale contains questions related to family, social, and economic status. Disease severity was also measured for both COPD and heart failure patients. However, by different questionnaires. Their QoL was measured as well (EQ-5D and WHOQOL). Nevertheless, we did not want to explore these results in the current manuscript which is already too dense. Hence we wanted to test, whether the improved UX itself, independent of the patient’s status, can provide an increase in their adherence. 
  3. Thank you for that remark. We tried to condense the content and increase the readability of the provided text. Figures have been merged and text was restructured and rewritten. We believe that now we have been able to highlight the main points much better than in the original text. 
  4. We have expanded the discussion and provided deeper analyses of all acquired results. 

We look forward to hearing from you regarding our submission and are ready to respond to your further questions and remarks.

On behalf of the author team

Ladislav Stanke and Ondrej Gergely

Reviewer 2 Report

Comments and Suggestions for Authors

General comment: This article outlines a step-by-step procedure to developing user experience (UX) for an mHealth application targeting older adult patients with chronic diseases, incorporating both quantitative and qualitative methods. This topic is current, intriguing, and highlights the complexity of UX development in mHealth applications for older adults with chronic diseases, underscoring the importance of addressing technical challenges for user satisfaction and engagement. While at times challenging, the article provided an engaging read. I encourage the authors to consider my feedback for future iterations of this manuscript. In the subsequent sections, I elaborate on each rating in my report.

Is the content succinctly described and contextualized with respect to previous and present theoretical background and empirical research (if applicable) on the topic? Must be improved

There is no background section. However authors succinctly introduce related work pieces (HF, COPD, medication adherence, and UX) in the introduction. Specifically, within the UX subsection, they present two research instruments, the User Experience Questionnaire (UEQ) and the System Usability 38 (SUS). However, other instruments such as the Medication Adherence Report Scale (MARS) questionnaire and the Beliefs about Medicines Questionnaire (BMQ) used in the research are not presented in the paper. This absence complicates the understanding of certain results, particularly for readers without a medical background, or necessitates independent search efforts. Conversely, ISO 14915 and ISO 9241 receive extensive coverage in the intro UX subsection (pp. 4-5) without explanation regarding their relevance to the research. While ISO 14915 is referenced again later in the paper (p. 9, 18, 21, and 23), its explicit application or relevance remains unclear, leaving readers to infer its use themselves. Similarly, ISO 9241 is not referenced beyond page 5, leaving readers to deduce its relevance themselves, thus obscuring its utilization in the research.

In my opinion, all instruments including ISO standards belong to the method section, and should receive even coverage in a “method” section where their relevance for the research should be made explicit. Finally, some usability testing studies are quantitative (execution time, success rate, number of clicks, etc.). To categorize usability testing solely as a qualitative method overlooks its quantitative dimensions and is therefore a misconception.

Are all the cited references relevant to the research? Yes

Are the research design, questions, hypotheses and methods clearly stated? Must be improved

The current structure of “2. Materials and methods” section, encompassing five steps, lacks clarity regarding the interconnections between these steps. The absence of a clear overview of the methods and instruments employed in each step contributes to a cumbersome reading experience, requiring readers to sift through extensive text and decipher the intricacies of every sentence. This unstructured presentation diminishes readability and comprehension, leading to a tedious and potentially confusing experience for readers. Therefore, a well-structured approach with explicit delineation of each step, its objectives, methods, instruments, and their relevance to the research, is essential for enhancing clarity, coherence, and reader engagement.

To enhance the clarity and structure of the methods section, I suggest renaming it to "Approach" or “Step-by-step procedure”, offering a concise overview of the research process, its five steps, and the interconnections between them. In addition, the evolution of the system should be presented here in a concise manner. Each step should then be delineated into its own section with a consistent structure:

  1. Specific objective: Clearly state the objective of the step.
  2. Methods and instruments: Detail the methods employed and the instruments utilized in this step.
  3. Scenario/Task (if applicable for usability testing): Describe the scenario or task used.
  4. Participants: Outline the number and characteristics of the participants involved in this step, the way they were recruited, etc.
  5. Implementation/Execution: Present the way the step was carried out, including the practical application of methods and instruments.
  6. Results: Summarize the findings or outcomes of this step.
  7. Discussion and integration with next step: Discuss results and explain how the findings of this step contribute to or influence the subsequent step.

Are the arguments and discussion of findings coherent, balanced and compelling? Must be improved

The current presentation of results lacks discussion of the findings, resembling a technical report with redesign solutions but little analysis of the data that were collected. The paper's structure complicates the identification of arguments and discussion. I couldn't locate them. To address this, as suggested above, I recommend in each step summarizing results (data collection and analysis) and then explicitly discussing them.

For empirical research, are the results clearly presented? Must be improved

The clarity of presenting empirical research results needs improvement for several reasons. As previously mentioned, the presentation lacks discussion, making it akin to a technical report with limited analysis. Additionally, most figures suffer from blurriness and lack of resolution, hindering their intelligibility. To enhance readability, UEQ and SUS results should be consolidated into single figures or tables, currently they spread on over 15 pages. Furthermore, the narrative surrounding (re)design is overly verbose and would benefit from synthesis within the 'Approach' or 'Step-by-step procedure' sections, with further discussion in the 'Discussion and integration with next step' section, as previously suggested.

Is the article adequately referenced? Yes

Are the conclusions thoroughly supported by the results presented in the article or referenced in secondary literature? Can be improved

The conclusions presented in the article have room for improvement. While the authors assert a mixed approach, the predominant focus on statistical analysis may not be suitable given the limited sample size. I recommend shifting the narrative emphasis away from statistical differences and towards other pertinent aspects of the research. By doing so, the authors can provide a more comprehensive and balanced discussion of their findings, incorporating both quantitative evidence and qualitative insights.

Comments on the Quality of English Language

Some typos here and there, but nothing particular to report here. 

Author Response

Dear reviewer,

thank you very much for reviewing our manuscript and giving us insightful ideas for further improvement. We sincerely appreciate the feedback that you have given to us.

Based on your comments we have been able to substantially improve the manuscript.

On behalf of the co-authors, we would like to respond to your comments and ideas:

  1. We have added descriptions of BMQ and MARS tools that we have used. Truly, the ISO section was enormously lengthy, therefore, it was made brief and clarified. As for quantitative we have meant the use of standardized questionnaires. You are correct, that we could directly measure the users' behavior within the application utilizing other quantitative techniques, rather than by questionnaires. We believe that this is not a misconception, rather we are both speaking about quantitative methods, however, their data sources and their collection are conducted differently.
  2. We appreciate a lot your suggestion for enhancing the structure. We tried to follow your idea as we think that it increases substantially the readability of the manuscript. We hope we were able to apply this structure in the best possible way.
  3. We agree that the discussion was not elaborated enough. The results and discussion section have been enhanced thoroughly. It is true, that the manuscript is based on a technical report, which originally lacked a context, that we wanted to add as it was already available to us and we felt it is worth sharing. We hope that by deepening the discussion we made our findings more usable for other developers and researchers.
  4. Figures have been improved and condensed.
  5. We have also enhanced the connection between the qualitative and quantitative aspects of the presented research.

We look forward to hearing from you regarding our submission and are ready to respond to your further questions and remarks.

On behalf of the author team

Ladislav Stanke and Ondrej Gergely

Reviewer 3 Report

Comments and Suggestions for Authors

General comments

This paper aims to study and evaluate Patients' UX impact on medication adherence using quantitative and qualitative components measurements. This is an important and interesting topic. I have some concerns and suggestions that may help improve the paper.

Major comments

  1. This article needs to be shorter, and the author is advised to remove less relevant content, such as P4-6 on ISO.
  2. In addition, P6-8 may be considered as an Appendix to highlight the focus of this article.
  3. P27-42, Figures 12 to 26 can be considered to be reduced, and the test of individual variables can be done in a narrative paragraphs way that makes it easier for the reader to understand
  4. P45-46, it is recommended that the text description is sufficient for the testing of various statistical variables.
  5. And it is recommended that the authors consider that the main t-test and regression test are not significant, which means that the selected variables or hypotheses are in doubt, and this part should be explained in detail.

Minor comments

  1. More background should connect the quantitative and qualitative components measurements to medication adherence. How do the two methods differ in assessing adherence?
  2. Separate the results, implications for managers and future research to improve readability. Consider analyzing health literacy, risk factors, policies, reports, and frameworks.

Author Response

Dear reviewer,

thank you very much for reviewing our manuscript and giving us insightful ideas for further improvement. We sincerely appreciate the feedback that you have given to us.

Based on your comments we have been able to substantially improve the manuscript.

On behalf of the co-authors, we would like to respond to your comments and ideas:

  1. We agree, that the section on ISO standards was omitted, as the readers can refer to the respective ISO standards for detailed information.
  2. We have restructured the text, so we think that it is more comprehensive and it is not necessary to move it to Appendix.
  3. We agree that the amount of the Figures needs to be reduced. Hence, we summarized the UEQ Dimensions in a single figure (Fig. 12). Figure 13 (former Fig. 20) summarizes the UEQ Overall, Necessity, and Concerns (including the NCD). 
  4. Descriptions were improved, and tables were condensed.
  5. We have tried to thoroughly describe and discuss the results in the discussion section. This part has been substantially enhanced to provide insight into that matter. The main goal of this research was to find out, whether these variables can provide us with any insight into adherence. The self-reported adherence measurement techniques might not be sufficient, as the initial adherence, as reported by patients, was already high. This is now discussed in deeper detail. 
  6. Rather than difference the qualitative method here played firstly the role of exploratory tool and in the second stage the role of explanatory tool. We believe that now it should be less obscured.
  7. The participants‘ education level was only measured by the single-item questionnaire assessing what is confidence in filling out the health-related forms. Regarding social support, the Gijón scale was administered. The Gijón scale contains questions related to family, social, and economic status. Disease severity was also measured for both COPD and heart failure patients. However, by different questionnaires. Their QoL was measured as well (EQ-5D and WHOQOL). Nevertheless, we did not want to explore these results in the current manuscript which is already too dense. Hence we wanted to test, whether the improved UX only, independent of the patient’s status and other aspects, can provide an increase in their adherence. 

We look forward to hearing from you regarding our submission and are ready to respond to your further questions and remarks.

On behalf of the author team

Ladislav Stanke and Ondrej Gergely

Round 2

Reviewer 1 Report

Comments and Suggestions for Authors

The paper's aim is to monitor heart failure patients using digital technologies in a home environment, with a focus on verifying the benefits of telemedicine applications for data acquisition, aggregation, and transmission to a server. The project utilizes devices such as a scale, mobile EKG, smartwatch, blood pressure monitor, and oximeter, all capable of sending data via Bluetooth to a tablet. The main contributions of the paper include the development and testing of a telemedicine application (Medimonitor) for remote patient monitoring, the use of modern digital technologies to support active aging, and the potential for early detection of decompensation through algorithms or neural networks. The strengths of the project include its innovative approach to elderly care, the use of CE-marked or FDA-approved devices, and the minimization of risks through protective circuits and proper guidance for users.
Overall, it provides valuable insights and practical cases for the field of telemedicine. Therefore, based on its innovativeness, practicality, and potential contribution to the well-being of elderly patients, I agree with the publication of this article and look forward to seeing more research and results from the SHAPES project.

Author Response

Dear reviewer,

thank you very much for your positive words! We appreciate that very much.

On behalf of the author team

Ladislav Stanke

Reviewer 3 Report

Comments and Suggestions for Authors

With this revision, this article can be published directly.

Author Response

(The authors gave the same response as above.)
